

# Reviews and syntheses: Four Decades of Modeling Methane Cycling in Terrestrial Ecosystems

Xiaofeng Xu[1, 2. 3], Fengming Yuan[4], Paul J. Hanson[4], Stan D.Wullschleger[4], Peter E. Thornton[4], William J. Riley[5], Xia Song[1, 3], David E. Graham[6], Changchun Song[2], and Hanqin Tian[7]

1. Biology Department, San Diego State University, San Diego, CA, USA
2. Northeast Institute of Geography and Agro-ecology, Chinese Academy of Sciences, Changchun, Jilin, China
3. Department of Biological Sciences, University of Texas at El Paso, El Paso, TX, USA
4. Climate Change Science Institute and Environmental Sciences Division, Oak Ridge National
Laboratory, Oak Ridge, TN, USA
5. Earth Sciences Division, Lawrence Berkeley National Laboratory, Berkeley, CA, USA
6. Biosciences Division, Oak Ridge National Laboratory, Oak Ridge, TN, USA
7. International Center for Climate and Global Change Research, School of Forestry and Wildlife Sciences, Auburn University, Auburn, AL, USA.

*Correspondence to*: Xiaofeng Xu (xxu@mail.sdsu.edu)

## Abstract

Over the past four decades, a number of numerical models have been developed to quantify the magnitude, investigate the spatial and temporal variations, and understand the underlying mechanisms and environmental controls of methane ($CH_4$) fluxes within terrestrial ecosystems. These $CH_4$ models

are also used for integrating multi-scale $CH_4$ data, such as laboratory-based incubation and molecular analysis, field observational experiments, remote sensing, and aircraft-based measurements across a variety of terrestrial ecosystems. Here we summarize 39 terrestrial $CH_4$ models to characterize their strengths and weaknesses and to suggest a roadmap for future model improvement and application. Our key findings are that: (1) the focus of $CH_4$ models has shifted from theoretical to site- and regional-level

applications over the past four decades, (2) large discrepancies exist among models in terms of representing $CH_4$ processes and their environmental controls, and (3) significant data-model and model-model mismatches are partially attributed to different representations of landscape characterization and inundation dynamics. Three areas for future improvements and applications of terrestrial $CH_4$ models are: (1) $CH_4$ models should more explicitly represent the mechanisms underlying land-atmosphere $CH_4$





exchange, with emphasis on improving and validating individual $CH_4$ processes over depth and
horizontal space, (2) models should be developed that are capable of simulating $CH_4$ emissions across
highly heterogeneous spatial and temporal scales, particularly hot moments and hot spots, and (3)
efforts should be invested to develop model benchmarking frameworks that can easily be used for
model improvement, evaluation, and integration with data from molecular to global scales.

## 1. Introduction

Methane ($CH_4$) has ~26 times the warming potential of carbon dioxide ($CO_2$) over a 100-year
horizon and is the second most important anthropogenic greenhouse gas, accounting for ~15% of
anthropogenic forcing to climate change (Forster et al., 2007; IPCC, 2013; Rodhe, 1990). Therefore, an
accurate estimate of $CH_4$ exchanges between land and the atmosphere is fundamental for understanding
climate change (Bridgham et al., 2013; Nazaries et al., 2013; Spahni et al., 2011). The ecosystem
modeling approach has been one of the most broadly used integrative tools for examining mechanistic
processes, quantifying the budget of $CH_4$ flux across spatial and temporal scales (Arah and Kirk, 2000;
Arah and Stephen, 1998; Cao et al., 1995; Curry, 2007; Fung et al., 1991; Huang et al., 1998b; Nouchi
et al., 1994; Potter, 1997; Riley et al., 2011; Walter et al., 1996; Xu et al., 2007; Zhuang et al., 2004),
and predicting future flux (Anisimov, 2007). Specifically, many $CH_4$ models have been developed to
integrate data, improve process understanding, quantify budgets, and project exchanges with the
atmosphere under a changing climate (Cao et al., 1995; Grant, 1998; Huang et al., 1998a; Potter, 1997;
Riley et al., 2011; Tian et al., 2010; Zhuang et al., 2004). In addition, model sensitivity analyses help to
design field and laboratory experiments by identifying the most uncertain processes and parameters in
the models (Massman et al., 1997; Xu, 2010).

Based on the complexity of the $CH_4$ processes represented, $CH_4$ models fall into two broad
categories: (1) empirical models to estimate and extrapolate measured methanogenesis, methanotrophy,
or $CH_4$ emission at plot, country, or continental scales (Christensen et al., 1996; Eliseev et al., 2008;
Mokhov et al., 2007; Wania et al., 2010, 2009); and (2) process-based models used for prognostic



understanding of individual $CH_4$ processes in response to multiple environmental drivers and budget quantification (reviewed below).

Over the past decades, many empirical and process-based models have been developed, for example CASA (Potter, 1997), CH4MOD (Huang et al., 1998b), CLM4Me (Riley et al., 2011), DAYCENT (Del Grosso et al., 2000), DLEM (Tian et al., 2010; Xu and Tian, 2012), DNDC (Li, 2000a), *ecosys* (Grant, 1998), HH (Cresto-Aleina et al., 2015), MEM (Cao et al., 1995), TEM (Zhuang et al., 2004), etc. However, recent analyses and model inter-comparisons have shown that most of these models poorly reproduce regional- to global-scale observations (Bohn and Lettenmaier, 2010; Bohn et al., 2015; Melton et al., 2013; Wania et al., 2013). A comprehensive synthesis and evaluation of the mechanisms incorporated into these models is lacking. In this paper we summarize $CH_4$ models published over the past four decades, their evolution in terms of process representation, and their coupling with Earth System Models. We pay special attention to the key processes in $CH_4$ cycling, specifically $CH_4$ substrate cycling, methanogenesis, methanotrophy, and transport in the soil profile, and their environmental controls. Emphasis is given to how these mechanisms were simulated in various models and how they were categorized in terms of complexity and ecosystem function. Models for understanding reactions in bioreactors (Bhadra et al., 1984; Pareek et al., 1999), mining plots (De Visscher and Van Cleemput, 2003), and marine systems (Elliott et al., 2011) were excluded.

## 2. Primary $CH_4$ Processes

Biological methane production in sediments was first noted in the late 18[th] century (Wolfe, 2004), and the microbial oxidation of methane was proposed at the beginning of the 20[th] century (Hanson and Hanson, 1996). Since then, methane cycling processes have been intensively studied and documented (Christensen et al., 1996; Hakemian and Rosenzweig, 2007; Lai, 2009; Melloh and Crill, 1996; Mer and Roger, 2001), and most have been described mathematically and incorporated into ecosystem models (Table 1). Herein, we do not attempt to review all $CH_4$ processes, as a number of reviews have been published on this topic (Barlett and Harriss, 1993; Blodau, 2002; Bridgham et al., 2013; Cai, 2012; Chen et al., 2012; Conrad, 1995; Conrad, 1996; Hakemian and Rosenzweig, 2007; Higgins et al., 1981; Lai, 2009; Monechi et al., 2007; Segers, 1998; Wahlen, 1993). Rather, we focus on



primary $CH_4$ processes in terrestrial ecosystems, and their environmental controls from a modeling perspective. In this context there exist three major methanogenesis mechanisms, two $CH_4$ methanotrophy mechanisms, and (depending on how one counts) seven $CH_4$ transport pathways in plants and soils. We note that most models do not explicitly represent all of these transport pathways, and that the relative importance of these pathways varies substantially in time and space. We also pay attention to several other modeling features including capability for plot- or regional-level simulations, vertical representation of biogeochemical processes, and whether the model is embedded in an Earth System Model.

The published literature concludes that two processes dominate biological $CH_4$ production (Conrad, 1999; Krüger et al., 2001): acetoclastic and hydrogenotrophic methanogenesis, which account for ~50% - 90% and ~10% - 43% of global annual $CH_4$ produced, respectively (Conrad and Klose, 1999; Kotsyurbenko et al., 2004; Mer and Roger, 2001; Summons et al., 1998). Methylotrophic methanogenesis (producing $CH_4$ from methanol, methylamines, or dimethylsulfide) is usually considered a minor contributor of $CH_4$, but may be significant in marine systems (Summons et al., 1998). The proportion of $CH_4$ produced via any of these pathways varies widely in time, space, and across ecosystem types.

Methanotrophy occurs under aerobic (Gerard and Chanton 1993) and anaerobic (Smemo and Yavitt 2011) conditions. These oxidative processes can occur in several locations in soil and plants (Frenzel and Rudolph 1998, Heilman and Carlton 2001, Ström et al. 2005) and using $CH_4$ either produced in the soil column or transported from the atmosphere (Mau et al. 2013). Large variation in the relative magnitudes of these pathways as a percentage of total methanotrophy has been observed: aerobic oxidation of $CH_4$ in soil contributes 1% - 90% (King, 1996; Ström et al., 2005), anaerobic oxidation of $CH_4$ within the soil profile contributes 0.3% - 5% (Blazewicz et al., 2012; Murase and Kimura, 1996), oxidation of $CH_4$ during transport in plant aerenchyma contributes <1% (Frenzel and Karofeld, 2000; Frenzel and Rudolph, 1998), and oxidation of atmospheric $CH_4$ contributes $\sim 10 - 100\%$ (~10% for wetland and ~100% for upland) (Gulledge and Schimel, 1998a; Gulledge and Schimel, 1998b; Topp and Pattey, 1997) to total methanotrophy. $CH_4$ is transported from the soil profile to the





atmosphere in typical open-water wetlands by several pathways: diffusive and advective plant-mediated transport accounts for 12~98% (Butterbach-Bahl et al., 1997; Mer and Roger, 2001; Morrissey and Livingston, 1992), soil gaseous diffusion accounts for ~5% for wetlands and > 90% for upland systems (Barber et al., 1988; Mer and Roger, 2001) (soil gaseous advection and aqueous diffusion and advection are typically relatively small (Tang et al., 2013), and ebullition accounts for 10~60% (Chanton et al.,

1989; Tokida et al., 2007) of the $CH_4$ produced in the soil that is emitted to the atmosphere. These processes vary significantly, depending on temporal and spatial scales.

Environmental factors affecting $CH_4$ processes have many direct and indirect controls. The dominant direct factors controlling methanogenesis and methanotrophy in most systems include oxygen availability, dissolved organic carbon concentration, soil pH, soil temperature, soil moisture, nitrate and

other reducers, ferric iron, microbial community structure, active microbial biomass, wind speed (Askaer et al. 2011), plant root structure (Nouchi et al. 1990), etc. Indirect factors include soil texture and mineralogy, vegetation, air temperature, soil fauna, nitrogen input, irrigation, agricultural practices, sulfate reduction, and carbon quality, etc. (Banger et al., 2012; Bridgham et al., 2013; Hanson and Hanson, 1996; Higgins et al., 1981; Mer and Roger, 2001). The complicated effects induced by a few

key factors on $CH_4$ processes have been mathematically described and incorporated in many $CH_4$ models; for example, direct factors such as temperature, moisture, oxygen availability, soil pH, and soil redox potential (Grant, 1998; Riley et al., 2011; Tian et al., 2010; Zhuang et al., 2004). The indirect factors such as nitrogen input (Banger et al., 2012), irrigation (Wassmann et al., 2000), and agricultural practices were not reviewed in this study as their impacts are indirect and were modeled through

impacts on vegetation and hydrology (Li, 2000a; Ren et al., 2011; Xu et al., 2010).

## 3. Model Representation of $CH_4$ Processes

*[Insert Figure 1 here]*

We reviewed 39 $CH_4$ models (Fig. 1 & Table 1), which were developed for a variety of purposes. The first $CH_4$ model was published in 1986 by Lovley & Klug (1986) to simulate

*methanogenesis* in water sediments, and since then a number of $CH_4$ models have been developed and applied at numerous scales (Table 1). For example, *Cao et al.* developed the Methane Emission Model



(MEM) and applied it to quantify the global $CH_4$ source in rice paddies and the sensitivity of the global $CH_4$ budget's response to climate change (Cao et al., 1995; Cao et al., 1998). Walter *et al.* developed and applied an ecosystem $CH_4$ model to quantify global $CH_4$ emission in response to water level

fluctuations (Walter and Heimann, 2000; Walter et al., 1996). Grant *et al* (1998) developed the *ecosys* model, which is currently the ecosystem-scale model that most mechanistically represents the many kinetic processes and microbial mechanisms for methanogenesis, methanotrophy, and $CH_4$ emission (Grant and Roulet, 2002). *Huang et al.* developed the CH4MOD model to investigate $CH_4$ processes and flux in rice paddies (Huang et al., 1998b; Huang et al., 2004; Li et al., 2012). Zhuang *et al*

developed a $CH_4$ module within the terrestrial ecosystem model (TEM) framework and applied it at site- and regional-levels (Zhuang et al., 2004; Zhuang et al., 2006), particularly across high latitudes (Fan et al., 2013; Zhu et al., 2011, 2013a, b, 2014). Tian et al. (2010) developed the dynamic land ecosystem model (DLEM), which is capable of simulating methanogenesis, methanotrophy, and $CH_4$ transport in terrestrial ecosystems; the scale of its application has ranged widely (i.e. plot, country,

continent, and global) for budget estimation and attribution analysis (Ren et al., 2011; Tian et al., 2015; Tian et al., 2011; Xu and Tian, 2012; Xu et al., 2010). Riley et al (2011) developed CLM4Me, a $CH_4$ module for the Community Land Model, which is incorporated in the Community Earth System Model. The family of LPJ models (LPJ-Bern, LPJ-WHyMe, LPJ-WSL) was developed under the LPJ framework to simulate $CH_4$ processes, but with different modules for $CH_4$ cycling; for example, LPJ-

Bern and LPJ-WHyMe incorporate Walter's $CH_4$ module (Walter and Heimann, 2000; Walter et al., 1996; Wania et al., 2009) while LPJ-WSL incorporates the $CH_4$ module from Christensen et al (Christensen et al., 1996).

*[Insert Figure 2 here]*

The number of $CH_4$ models has steadily increased since the 1980s (Figs. 1 & 2): 1 in the 1980s,

12 in the 1990s, 14 in the 2000s, and 12 for 2010-2015. This rapid increase in $CH_4$ model development indicates the rapidly growing effort to analyze $CH_4$ cycling and quantify $CH_4$ budgets across spatial scales. Meanwhile, the key mechanisms represented in the models have changed at a slower pace (Fig. 2). The most important changes are representation of vertically-resolved processes within the soil and



regional model simulation. For example, the percentage of the newly developed models with vertically-
resolved $CH_4$ biogeochemistry has increased from 54% before 2000 to ~83% in the recent decade
(2010-2015). The proportion of models with regional simulation capability has doubled from ~50%
before the 2010s to almost 100% afterwards (Fig. 2).

*[Insert Tables 1, 2, and 3 here]*

The majority of these models were designed to simulate land-surface exchanges in saturated
ecosystems (primarily natural wetlands and rice paddies) (Huang et al., 1998b; Li, 2000a; Walter et al.,
1996) (Table 1). Not all of the models explicitly represented the belowground mechanistic processes for
$CH_4$ production and consumption and the primary carbon biogeochemical processes (Christensen et al.,
1996; Ding and Wang, 1996). The land-atmosphere $CH_4$ exchange is a net balance of many processes
including production, oxidation, and transport, which are represented in models with different
complexities (Table 2). Some models are very complicated, while some are relatively simple. The
obvious tradeoff in modeling $CH_4$ cycling is to represent mechanisms as accurately as possible while
managing complexity (Evans et al., 2013), and ensuring that additional complexity enhances
predictability (Tang and Zhuang, 2008).

**4. $CH_4$ Model Classification**

80                          *[Insert Figure 3 here]*

Current $CH_4$ models can be classified into three groups based on their representation of primary
mechanistic processes for methanogenesis, methanotrophy (Fig. 3), and $CH_4$ transport (Fig. 4). The first
group of $CH_4$ models uses a very simple framework for land-surface $CH_4$ flux, and most were
developed before the 2000s (e.g., Christensen's model, CASA, etc.) (Fig 3A). These models treated
land-surface $CH_4$ flux as an empirical function and link it to environmental controls, or soil organic
carbon; this group of models ignored the mechanistic processes of methanogenesis, methanotrophy, and
$CH_4$ transport. The second group of $CH_4$ models considers processes in a relatively simple manner (e.g.,
one or two primary $CH_4$ transport pathways, methanogenesis as a function of DOC, oxidation of
atmospheric $CH_4$, etc.); however, the methanogenesis and methanotrophy mechanisms are still not
mechanistically represented (Fig. 3B). The third group of $CH_4$ models explicitly simulates the processes




for methanogenesis, methanotrophy, and CH$_4$ transport as well as their environmental controls, which allows comprehensive investigation of physical, chemical, or biological processes' contribution to land-surface CH$_4$ flux (Fig. 3C). Of the models in the third group, none of them fully represent all these processes (although some have most of the features described); for example, the *ecosys* model is one of

the few models to represent most of the CH$_4$ cycling processes shown in Fig. 3C, although it has not been embedded in an Earth System Model. We recommend that the third group of CH$_4$ models be the focus of mechanistic studies, and the basis for improving reduced form models applicable to Earth System Model applications.

## 4.1. Methanogenesis

Models make use of four types of modeling frameworks (Table 3) to relate methanogenesis to substrate requirements. Similar to Eqs (1) – (4) in Table 3, the model representation of methanogenesis can be classified into four types: (1) empirical association between methanogenesis and environmental condition, including temperature and water table; (2) empirical correlation of methanogenesis with biological variables (particularly heterotrophic respiration and soil organic matter); (3) methanogenesis

as a function of concentration of substrate (DOC); and (4) a suite of mechanistic processes simulated for methanogenesis.

Representation of the substrate for methanogenesis may be a key aspect of simulating CH$_4$ cycling in terrestrial ecosystems; however, more than half of the models we examined do not explicitly simulate substrates for methanogenesis. We note, however, that explicit representation of substrates and

their effects on methanogenesis requires additional model parameters, and therefore degrees of freedom in the model, which can lead to increased equifinality (Tang and Zhuang, 2008). The optimum complexity level for methanogenesis and consumption models remains to be determined.

The first group of models correlates methanogenesis with environmental factors and ignores substrate production and its influence on methanogenesis [Eq. (1)] (Table 3). This group of models

includes Christensen's model (Christensen et al., 1996), which simulates the net flux of CH$_4$ based on fraction of saturated soil column and soil temperature, and the IAP-RAS model (Mokhov et al., 2007), which calculates methanogenesis as an empirical equation of soil temperature. This group of models has



a role in site-specific interpolation of observations for scaling over time at a given site, but does not explicitly represent carbon or acetate substrate.

The second group of models directly links methanogenesis with heterotrophic respiration or soil organic matter content, but does not explicitly represent carbon or acetate substrate availability [Eq. (2)]; examples are the LPJ model family (Hodson et al., 2011; Spahni et al., 2011; Wania et al., 2010, 2009) and CLM4Me (Riley et al., 2011).

      The third group of models simulates dissolved organic carbon (DOC) or different pools of soil

organic carbon, which are treated as a substrate pool influencing $CH_4$ production [Eq. (3)]; examples are the MEM model (Cao et al., 1995; Cao et al., 1998) and DLEM (Tian et al., 2010).

      The fourth group of $CH_4$ models considers the primary substrates for methanogenesis, that is, acetate and single-carbon compounds [Eq. (4)]; examples are Kettunen's model (Kettunen, 2003), Segers' model (Segers and Kengen, 1998; Segers and Leffelaar, 2001a, b; Segers et al., 2001), van

Bodegom's model (van Bodegom et al., 2000; Van Bodegom et al., 2001), and the *ecosys* model (Grant, 1998).

      Methanogenesis is a fundamental process for $CH_4$ cycling, and a majority of models simulate methanogenesis in either implicit or explicit ways (Tables 2 & 3). For example, 31 models (i.e. Arah's model, "Cartoon" model, CASA, CH4MOD, Christensen's model, CLM4Me, Ding's model, DLEM,

DNDC, ecosys, Gong's model, HH model, IAP-RAS, Kettunen's model, Lovley's model, LPJ-Brn, LPJ-WHyMe, LPJ-WSL, Martens' model, MEM, MERES, ORCHIDEE, SDGVM, Segers' model, TEM, TRIPLEX-GHG, UW-VIC, van Bodegom's model, VISIT, Walter's model, and Xu's model) simulate methanogenesis as one individual process. As a comparison, only three out of 39 $CH_4$ models reviewed explicitly simulate two methanogenesis pathways (acetoclastic methanogenesis and

hydrogenotrophic methanogenesis) (Table 3). As mentioned earlier, it is well-recognized that there are two dominant methanogenesis pathways and their relative combination changes significantly across environmental gradients, for example, along the soil profile (Falz et al., 1999) and across landscape types (McCalley et al., 2014). This lack of representation of two methanogenesis mechanisms might have caused dramatic bias and needs to be address in future model improvements.



## 4.2. Methanotrophy

Methanotrophy is another important process for simulating the land-atmosphere exchange of $CH_4$ (Table 2). Aerobic and anaerobic methanotrophy occurs in different locations in the soil profile, and affect both methanogenesis in the profile and $CH_4$ diffusing in from the atmosphere. For example, the oxidation of atmospheric $CH_4$, rhizosphere and bulk soil oxidation, and oxidation during $CH_4$ transport from soil to the atmosphere have been measured and modeled (Tables 1 & 2). Anaerobic $CH_4$ oxidation has been measured (Blazewicz et al., 2012) and has proposed to be incorporated into ecosystem models (Gauthier et al., 2015).

It has been confirmed that the aerobic oxidation of $CH_4$ produced in the soil profile and aerobic oxidation of atmospheric $CH_4$ play a major role in $CH_4$ consumption in the system, and that anaerobic oxidation of $CH_4$ is a minor contributor. Currently, no models explicitly simulate the anaerobic oxidation of $CH_4$ in soil, although a few recent studies highlighted the importance of this process (Blazewicz et al., 2012; Caldwell et al., 2008; Conrad, 2009; Smemo and Yavitt, 2011; Valentine and Reeburgh, 2000). The key reasons for this omission are that the process has not been mathematically described, the key parameters are uncertain (Gauthier et al., 2015), and the biochemical mechanism is not fully understood.

The Michaelis-Menten-like equations, widely used for simulating $CH_4$ production and oxidation, consider substrates limiting factors (Segers and Kengen, 1998). A few $CH_4$ models in the third category (linking methanogenesis with a substrate) use the Michaelis-Menten-like equation to compute methanogenesis and methanotrophy rates (Eqs. 3, 5, & 6). For example, DLEM simulates methanogenesis as a function of DOC concentration and other environmental controls, and Michaelis-Menten-like functions were used to compute methanogenesis on the basis of DOC as substrate. Methanotrophy has been simulated with dual Monod Michaelis-Menten-like equations with $CH_4$ and oxygen as limiting factors (Table 3). We note that the Michaelis-Menten-like relations may be inaccurate when representing multi-substrate, multi-consumer networks (Tang and Riley 2013, 2015). Although their approach (Equilibrium Chemistry Approximation, ECA) has not been applied for simulations of $CH_4$ emissions, $CH_4$ dynamics are inherently multi-consumer, including transformations





associated with methanogens, heterotrophs, ebullition, advection, diffusion, and aerenchyma transport, even if only one substrate is considered.

### 4.3. $CH_4$ within the Soil/Water Profile

$CH_4$ produced in the soil profile or below the water table is not transported immediately into the atmosphere. The time required for $CH_4$ to migrate from deep soil profile to the atmosphere ranges from minutes to days (depending on temperature, water, soil texture, and emissivity of plant roots), or even a season if the surface is frozen. The majority of current $CH_4$ models assume that $CH_4$ transport to the atmosphere occurs immediately after $CH_4$ is produced, and a portion is oxidized (Tian et al., 2010;

Zhuang et al., 2004); for models simulating $CH_4$ flux over minutes to days, the lack of modeled transport may produce unrealistic simulations.

Some models do simulate $CH_4$ dynamics within the soil and water profile (e.g., ecosys, CLM4Me), which produces a lag between methanogenesis and emission, allowing for oxidation to be explicitly represented during transport, and is valuable for simulating the seasonality of $CH_4$ flux (Table

2). For example, the recently observed $CH_4$ burst in the spring season in some field experiments confirms that the storage of $CH_4$ produced in winter will likely produce a strong emission outburst (Song et al., 2012). Without the mechanism of $CH_4$ storage beneath the soil surface, this phenomenon is impossible to simulate. In most of the models considering $CH_4$ storage, the $CH_4$ is treated as a simple gas pool, under the water table, which will be transported to the atmosphere through several transport

pathways.

### 4.4. $CH_4$ Transport from Soil to the Atmosphere

The transport of $CH_4$ produced and stored in soil column is the final bottleneck for $CH_4$ leaving the system; therefore, this process is an important control on the instantaneous land-surface $CH_4$ flux. Several important pathways of $CH_4$ transport to the atmosphere are identified: plant-mediated diffusive

and advective transport, aqueous and gaseous diffusion, and ebullition (Beckett et al., 2001; Chanton, 2005; Mer and Roger, 2001; Whiting and Chanton, 1996). Model simulation of these transport pathways uses direct control of simulated land surface $CH_4$ flux, with $CH_4$ transport simulation considered in a manner similar to Eq. (7) (Table 3).



The majority (77%) of the current models simulate at least one transport pathway. Specifically, 66% of the models simulate $CH_4$ transport via aerenchyma, 77% simulate gaseous diffusive transport, and 54% simulate ebullition transport (Table 1). More than 50% of models simulated these three transport pathways. Some models simulate explicitly the aqueous and gaseous diffusion of $CH_4$ (Riley et al., 2011), while most models do not simulate advective transport. Many models simulate diffusion and plant-mediated transport in very simple ways. For model improvement in this area, three issues remain as challenges:

(1)     Most models treat transport implicitly; for example, the diffusion processes is treated simply as an excessive release of $CH_4$ when its concentration exceeds a threshold (Tian et al., 2010). This treatment prevents the model from simulating the lag between methanogenesis and its final release to the atmosphere, which has been confirmed to be the key mechanism for hot-moment and hot-spot of $CH_4$ flux (Song et al., 2012) and for oxidation during transport.

(2)     The parameters for plant species capable of transporting gas (i.e., *aerenchyma*) are poorly constrained (Riley et al. 2011), although plant-mediated transport has been identified as the dominant pathway for $CH_4$ emission in most natural wetlands (Aulakh et al., 2000; Colmer, 2003).

(3)     Simultaneously representing aqueous and gaseous phases of $CH_4$ is one potentially important issue for simulating $CH_4$ transport from soil to the atmosphere (Tang and Riley, 2014). However, these processes are only explicitly represented in a few extant $CH_4$ models (Riley et al., 2011; Grant et al., 1998).

### 4.5. Environmental Controls on $CH_4$ Processes

Although a suite of environmental factors affects various $CH_4$ processes, many of these factors are not explicitly simulated in many models. These factors include soil temperature, soil moisture, substrate, soil pH, soil redox potential, and oxygen availability. Many other factors not directly incorporated in the models, could indirectly affect $CH_4$ cycling. For example, nitrogen fertilizer affects methanogenesis through its stimulating impacts on ecosystem productivity, which in turn affects DOC, soil moisture and soil temperature (Xu et al., 2010). The CLM4Me model simulates permafrost and its



effects on CH$_4$ dynamics, and has a simple relationship for soil pH impacts on methanogenesis (Riley et al., 2011). In this review, we specifically focus on temperature, moisture, and pH because these factors directly affect CH$_4$ processes in all environments, and they have been explicitly simulated in the many of the models.

Three types of mathematical functions have been used to simulate the temperature dependence of CH$_4$ processes: (1) linear functions of air or soil temperature (Eq. 10 in Table 3), (2) Q$_{10}$ function (Eq. 10 in Table 4), and (3) Arrhenius type function (Eq. 12 in Table 3). Of these three model representations of temperature dependence, the Q$_{10}$ equation is the most common mathematical description. However, the parameters for these empirical functions vary widely across the models

(Table 4). Actual temperature responses may diverge significantly from the models at low temperatures, close to the freezing point of water, and high temperatures, close to the denaturation point of enzymes.

*[Insert Table 4 here]*

Soil moisture is another important factor controlling CH$_4$ processes, because water limits O$_2$ diffusion from the air through the soil column and because microbes can become stressed at low matric

potential. CH$_4$ is produced typically under conditions with a low reduction potential, which is normally associated with long-term inundation. Although methanogenesis occurs solely under reducing conditions (methanogenesis within plant biomass under aerobic condition has never been simulated although it has been reported in experiments (Keppler et al., 2006)), methanotrophy occurs under drier, aerobic conditions. A low water content can also limit microbial activity in frozen soils or soils with

high osmolarity (Watanabe and Ito, 2008). Therefore, soil moisture has different impacts on different CH$_4$ processes. Four types of model representation are used to simulate moisture's effects on CH$_4$ processes (Eqs. 12-15 in Table 3).

(1) Methanogenesis occurs only in the saturated zone and an exponential function for soil moisture is used to control methanotrophy (e.g., CLM4Me);

(2) Linear function for moisture impacts (e.g., CLASS use linear function for moisture impact on methanotrophy) (Curry, 2007);





(3) Reciprocal responsive curves for moisture impacts on methanogenesis and methanotrophy (e.g., DLEM) (Tian et al., 2010);

(4) A bell-shaped curve for methanogenesis (e.g., TEM uses a function similar to Eq. (16) for moisture impacts) (Zhuang et al., 2004).

The pH is another important factor that has been included in a number of $CH_4$ models (Cao et al., 1995; Zhuang et al., 2004). Methanogens and methanotrophs depend on proton and sodium ion translocation for energy conservation, thus they are directly affected by pH. The pH impacts on $CH_4$ processes are simulated as a bell-shaped curve although the mathematical functions used to describe pH impacts are different (Eq. 17a, 17b, and 17c). Moreover, even when the same functions were used in different models, they were associated with different parameter values; for example, the MEM model sets $pH_{min}$ (minimum pH value for $CH_4$ processes being active), $pH_{opt}$ (optimal pH value for $CH_4$ processes being most active), and $pH_{max}$ (minimum pH value for $CH_4$ processes being active) values of 5.5, 7.5, and 9 (Cao et al., 1995). This set of parameter values was adopted in the TEM model (Zhuang et al., 2004), whereas the DLEM model uses values of 4, 7, and 10 (Tian et al., 2010). The CLM4Me model uses a different function while keeping the impact curve at the same shape, but its peak has an optimal pH of 6.2 (Meng et al., 2012).

For the other environmental factors, model representation is still in its infancy; however, several models consider oxygen availability as an electron acceptor for methanotrophy (e.g., Arah's model, Beckett's model, "Cartoon" model, CLM4Me, *ecosys*, Kettunen's model, MERES, Segers' model, van Bodegom's model, De Visscher's model, and Xu's model). In addition, only a few models simulate the impacts of the electron acceptor (i.e. nitrate, sulfate, etc.) on $CH_4$ processes (Table 2). For example, van Bodegom's model simulates iron biogeochemistry, and Lovley's, Marten's, and van Bodegom's models simulate sulfate as the electron acceptor and its impacts on methanogenesis and methanotrophy (Lovley and Klug, 1986; Martens et al., 1998; Van Bodegom et al., 2001). Explicitly representing these processes enables future coupling of $CH_4$ cycling to processes that are regionally significant, such as iron reduction on the Alaskan North Slope (Miller et al., 2015). These models' representation has the





advantage of more accurately simulating biogeochemical processes of carbon and ions, although large uncertainties still exist because of the lack of data for constraining model parameters.

**5. Summary**

Through the four decades of modeling $CH_4$ cycling in terrestrial ecosystems, consensus has been reached on several fronts. First, $CH_4$ cycling includes a suite of complicated processes, and both the simple and complex models are able to estimate land-surface $CH_4$ flux to a certain level of confidence, although models of different complexity do provide different results (Tang et al., 2010). Second,

although a number of $CH_4$ models have been developed, several gaps remain that need new model representations (e.g., dynamic linkage between inundation dynamics and the $CH_4$ module (Melton et al., 2013), anaerobic oxidation of $CH_4$ (Gauthier et al., 2015)).

Two recent $CH_4$ model-model inter-comparison projects raised several important points (Bohn et al., 2015; Melton et al., 2013): (1) the distribution of the inundation area is important for accurately

simulating global $CH_4$ emissions, but was poorly represented in $CH_4$ models; (2) the modeled response of land-surface $CH_4$ emission to elevated $CO_2$ is likely biased as a number of global change factors were missing, which indicates the need for modeling with multiple global environmental factors; and (3) the need for comparison with high-frequency observational data is identified as an important task for future model-model inter-comparison. These lessons will be helpful for, and likely addressed during,

model improvements and applications of more mechanistic $CH_4$ models.

Although the primary individual $CH_4$ processes have been studied and quantified at a certain level of confidence, only a few modeling studies have reported these individual processes. For example three pathways of $CH_4$ transports were represented in Kettunen, 2003 and Walter et al., 1996, but none of those modeled results have been evaluated against observational results for those individual

processes. One reason is that measurements rarely distinguish among individual processes; another reason is that the majority of $CH_4$ models do not explicitly represent all processes (Table 2). However, a number of studies report significant shifts in the processes contributing to the surface $CH_4$ flux along environmental gradients or across biomes (Conrad, 2009; Krumholz et al., 1995; McCalley et al., 2014). Projecting $CH_4$ fluxes into future changing climate conditions requires not only accurate simulations of




$CH_4$ processes, but also shifts among the various processes. In addition, $CO_2$ flux has been evaluated within the Earth System Modeling framework, but only a few studies have evaluated the $CH_4$ flux and its contribution to climate dynamics. Given the much higher warming potential and relatively faster rate of increase of atmospheric $CH_4$, fully coupled simulations are needed to represent the feedbacks between terrestrial $CH_4$ exchanges and climate. We note that a few recent studies reported a relatively

small climate warming-methane feedback from global wetlands and permafrost (Gao et al., 2013; Gedney et al., 2004; Riley et al., 2011). A fully mechanistic $CH_4$ model that accounts for all the important features is critically needed. In addition, a modeling framework to integrate multiple sources of data, such as microbial community structure and functional activities, ecosystem-level measurements, and global scale satellite measurements of gas concentration and flux is needed with

these mechanistic $CH_4$ models.

## 6. Needs for Mechanistic Methane Models

During the recent few years, the scientific community has continued to improve and optimize models to better simulate methanogenesis, methanotrophy, $CH_4$ transport, and their environmental and biological controls (Xu et al., 2015; Zhu. Q. et al., 2014). A number of emerging tasks have been

identified, and progress in these directions is expected. First, linking genomic data with large-scale $CH_4$ flux measurements will be an important, while challenging, task for the entire community; for example, some work has been carried out in this direction (De Haas et al., 2011; Larsen et al., 2012). An effort has been initialized to develop a new microbial functional group-based $CH_4$ model, which has the advantages of linking genomic information for each individual process with the four microbial

functional groups (Xu et al., 2015). Second, data-data and model-model comparisons are another important effort for model comparison and improvement. One ongoing encouraging feature that all recently developed $CH_4$ models possess is the capability for regional simulations as well as the possibility to be run at the site level (Riley et al., 2011; Zhu. Q. et al., 2014).

Third, microbial processes need to be considered for incorporation into ecosystem models for

simulating carbon cycling and $CH_4$ processes (DeLong et al., 2011; Xu et al., 2014). Although a few models explicitly simulate the microbial mechanisms of $CH_4$ cycling (Arah and Stephen, 1998; Grant,



1998; Li, 2000a; Segers and Kengen, 1998), none of them have been used for regional- or global-scale estimation of microbial contributions to the $CH_4$ budget. A reasonable experimental design and a well-validated microbial functional group-based $CH_4$ model should be combined to enhance our capability to

apply models to estimate a regional $CH_4$ budget and to investigate the combination of microbial and environmental contributions to the land surface $CH_4$ flux (DeLong et al., 2011). Fourth, incorporating well-validated $CH_4$ modules into Earth System Modeling frameworks will allow a fully coupled simulation that provides a holistic understanding of the $CH_4$ processes, with its connections to many other processes and mechanisms in the atmosphere. Several recently developed models fall in the

framework of Earth System Models (Riley et al., 2011; Ringeval et al., 2010), which provide a foundation for this application in a relatively easy way. This effort will likely contribute not only to the $CH_4$ modeling community, but also to the entire global change science community (Koven et al., 2011). The iron and sulfate biogeochemistry that has been simulated in a few models was not included in any of the three groups because that effort will likely be achieved over the long term, owing to poor

understanding of the mechanisms and the lack of observational data.

*[Insert Figure 4 here]*

Based on the above-mentioned needs and model features as well as the mechanisms for the $CH_4$ models, the next generation of $CH_4$ models will likely include several important features (Fig. 4). The models should (1) be embedded in an Earth System Model, (2) consider the vertical distribution of

thermal, hydrological, and biogeochemical transport and processes, (3) represent mechanistic processes for microbial $CH_4$ production, consumption, and transport, and (4) support data assimilation and a model benchmarking system as auxiliary components.

## 7. Challenges for Developing Mechanistic $CH_4$ Models

*Knowledge Gaps* - Modeling $CH_4$ cycling is a dynamic process. As new mechanisms are

identified the modeling community should ensure that the mechanisms are well studied and mathematically described, as has occurred over the past decades (Conrad, 1989; McCalley et al., 2014; Schütz et al., 1989; Xu et al., 2015). However, a number of knowledge gaps need to be filled before a full modeling framework of $CH_4$ processes within terrestrial ecosystems can be achieved. The first gap



is either confirmation or rejection of a few recently observed $CH_4$ mechanisms; these mechanisms need

to be fully vetted before being considered for incorporation into a model. The first most well-known

mechanism still under debate is aerobic $CH_4$ production within plant tissue (Beerling et al., 2008;

Keppler et al., 2006). Since its first report in 2006 (Keppler et al., 2006), a few studies have confirmed

the mechanism in multiple plant species (Wang et al., 2007). While its existence in nature is still under

debate (Dueck et al., 2007), this mechanism will likely not be incorporated into an ecosystem model

before solid evidence is presented and consensus is reached. The second new mechanism is fungi as a

microbial group carrying out $CH_4$ production (Lenhart et al., 2012). More field- or lab-based

experiments are needed to investigate this mechanism and its contribution to the global $CH_4$ budget,

probably through a data model integration approach. Third, the aerobic production of methane from the

cleavage of methylphosphonate has been demonstrated in marine systems (Karl et al., 2008), but the

significance of this process in terrestrial systems is unknown.

Another knowledge gap is the missing comprehensive understanding of spatial and temporal

variations in $CH_4$ flux; particularly, the "hot spots" and "hot moments" of observed $CH_4$ flux are still

not completely understood (Becker et al., 2008; Mastepanov et al., 2008; Song et al., 2012). The

traditional static chamber method of measuring $CH_4$ emissions could underestimate the $CH_4$ flux

because sparse sampling is unlikely to detect these foci or pulses of unusually high emissions. Better

methods are also needed to measure $CH_4$ cycling during the shoulder seasons in the Arctic and subarctic

when fluxes may be most variable (Zona et al. 2016). These knowledge gaps are key hurdles for $CH_4$

model development efforts. No model has yet been tested for simulating hot spots or hot moments over

large spatial or long temporal scales. However, the high range (usually of order 1-10) of these processes

might cause regional budgets to vary substantially (Song et al., 2012); therefore, mechanistic model

representations of these mechanisms are highly needed.

*Modeling Challenges* - Better simulation of $CH_4$ cycling in terrestrial ecosystems requires

improvement in the model structure to represent mechanistic $CH_4$ processes. First is the challenge for

better simulating the vertical profile of soil biogeochemical processes and validating these models with

observational results. Although some models have a capability for vertical distribution of carbon and



nitrogen (Koven et al., 2013; Tang et al. 2013; Mau et al., 2013), a better framework for $CH_4$ and extension to cover the majority of $CH_4$ models are needed. This vertical distribution of biogeochemistry is necessary for simulating the vertical distribution of $CH_4$ processes and $CH_4$ transport through the soil profile before reaching the atmosphere. A second challenge is incorporating tracer capability. Isotopic

tracers ($^{13}C$, $^{14}C$) have been widely used for quantifying the carbon flow and partitioning among individual $CH_4$ processes (Conrad, 2005; Conrad and Claus, 2005), but for ecosystem models this capability has not been represented even though it is very important to understanding $CH_4$ processes and integrating field observational data. A third challenge is to simulate microbial functional groups. Microbial processes are carried out by different functional groups of microbes (Lenhart et al., 2012;

McCalley et al., 2014). Therefore, model comparison with individual processes requires representing the microbial population sizes (or active biomass) for specific functional groups (Tveit et al., 2015). This goal has proved more difficult than representing plant functional types or traits in models, because not all microbial taxonomic groups have ecologically coherent functions (Philippot et al., 2010). A fourth challenge is to simulate the lateral transport of dissolved and particulate biogeochemical

variables that are necessary to better simulate the storage and transport of $CH_4$ within heterogeneous landscapes (Weller et al., 1995). A fifth challenge is modeling $CH_4$ flux across spatial scales. Although a few studies have been used to demonstrate the approach for simulating $CH_4$ budget at plot scale and eddy covariance domain scale (Zhang et al., 2012), a mechanistic framework to link $CH_4$ processes at distinct scales is still lacking while highly valuable.

*Data Needs* - First, a comprehensive dataset of field measurements of $CH_4$ fluxes across various landscape types is needed. Although a number of datasets have been compiled (Aronson and Helliker, 2010; Chen et al., 2012; Liu and Greaver, 2009; Mosier et al., 1997; Yvon-Durocher et al., 2014), some landscape types are still not fully covered. Meanwhile, high-frequency field observational data are also needed, particularly long-term observational data in some less-studied ecosystems. It is well-known that

inter-annual variation of climate may turn an ecosystem from a $CH_4$ sink to a $CH_4$ source (Nauta et al., 2015; Shoemaker et al., 2014); therefore, a long-term observational dataset that covers these shifts in $CH_4$ flux and its associated ecosystem information would improve our understanding of the processes




and our representation of them in $CH_4$ models. Second, microbial community shifts and their role in $CH_4$ processes are important, although information is incomplete for model representation of this

mechanism (McCalley et al., 2014; Schimel and Gulledge, 1998). Although a number of studies have reported the microbial community structure and its potential association with changes in $CH_4$ processes (Monday et al., 2014; Schimel, 1995; Wagner et al., 2005), none of this progress has been documented in a mathematical manner suitable for a modeling representation.

       Last but not least, a comprehensive dataset of all primary $CH_4$ processes within an individual

ecosystem would be valuable for model optimization and validation. Although some datasets exist, no study has investigated all primary individual $CH_4$ processes within the same plot over the long term. Given the substantial spatial heterogeneity of $CH_4$ processes, this lack of process representation may cause bias in $CH_4$ simulations at regional scale. It should be noted that land surface net $CH_4$ flux is a measurable ecosystem-level process, whereas many individual $CH_4$ processes are difficult to accurately

measure. Therefore, designing field- or lab-based-experiments suitable for measuring these processes is a fundamental need. For example, the anaerobic oxidation of $CH_4$ has been identified as a critical process for some ecosystem types, but no comprehensive dataset on it is available for model development or improvement.

       *Data-Model Integration* - Model development and data collection are two important, but

historically independent scientific approaches; the integration between model development and data collection is much stronger for advancing science (De Kauwe et al., 2014; Luo et al., 2012; Peng et al., 2011). Although data-model integration is recognized as very important for understanding and predicting $CH_4$ processes and some progress has been made, integrating experiments and models presents multiple challenges, particularly, 1) the methods for integrating data with the models are not

well developed for $CH_4$ cycling; 2) the metrics for evaluating data-model integration are not consistent in the scientific community; and 3) the regular communication between data scientists and modelers on various aspects of $CH_4$ processes and their model representation is lacking.

       Methods for data-model integration have been recently created, for example, Kalman Filter (Gao et al., 2011), Bayesian (Ogle and Barber, 2008; Ricciuto et al., 2008; Schleip et al., 2009; Van Oijen et



al., 2005), and Monte Carlo (Casella and Robert, 2005). However, no studies have evaluated these methods for integrating $CH_4$ data with models. In addition, the metric for evaluating the data-model integration is still not well developed. A very helpful strategy for data-model integration in to solicit timely input from modelers when designing a field experiment. A good example of this is the U.S. Department of Energy-sponsored project Next Generation Ecosystem Experiments - Arctic (ngee-arctic.ornl.gov), which was planned with inputs from field scientists, data scientists, and modelers. Another successful example is the U.S. DOE-sponsored project, SPRUCE (mnspruce.ornl.gov), in which the experiment design for data-model integration created an opportunity for modeling needs to be adopted by the field scientists.

## 8. Concluding Remarks

$CH_4$ dynamics in terrestrial ecosystems have been intensively studied, but model representation of $CH_4$ cycling has lagged. Currently, the primary mechanisms for $CH_4$ processes in terrestrial ecosystems are implicitly represented in many, but not all, ecosystem models. Development of $CH_4$ models began in the late 1980s, and the pace of growth has been fast since the 1990s. Model development shifted from theoretical analysis in the 1980s and 1990s to being more applied in the 2000s and 2010s, expressed as being more focused on regional $CH_4$ budget quantification and integration with multiple sources of observational data. Although some current $CH_4$ models consider most of the relevant mechanisms, none of them consider all the processes for methanogenesis, methanotrophy, $CH_4$ transport, and their primary environmental controls. Further, evidence demonstrating that incorporating all of these processes would lead to more accurate prediction is needed. Incorporating sophisticated parameter assimilation, uncertainty quantification, equifinality quantification, and metrics of the benefits associated with increased model complexity are therefore required.

The $CH_4$ models for accurate projection of land-climate feedback in the next few decades should: (1) use mechanistic formulations for primary $CH_4$ processes, (2) be embedded in Earth System Models for the global evaluation of terrestrial-climate feedback associated with $CH_4$ fluxes, (3) have the capacity to integrate multiple sources of data, which makes the model not only a prediction tool but also



an integrative tool, and (4) be developed in association with model benchmarking frameworks. These four characteristics pave the way for examining $CH_4$ processes and flux in the context of global change.



**Acknowledgements**:

The authors are grateful for financial and facility support from the University of Texas at El Paso. The authors are grateful for Dr. Yiqi Luo at University of Oklahoma for his comments on the manuscript. This review is part of the $CH_4$ modeling tasks within the NGEE-Arctic and SPRUCE projects sponsored by the US Department of Energy Office of Science. Contributions by FY, PJH, PET, SDW, and DEG are supported by the U.S. Department of Energy, Office of Science, Office of Biological and Environmental Research. Oak Ridge National Laboratory is managed by UT-Battelle, LLC, for the U.S. Department of Energy under contract DE-AC05-00OR22725. CS is supported by the National Natural Science Foundation of China (41125001), HT is supported by NASA Carbon Monitoring System Program (NNX14AO73G) and NASA Interdisciplinary Science Program (NNX14AF93G).




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





Table 1. Terrestrial ecosystem models for $CH_4$ cycling and the model representation of three pathways of $CH_4$ transport (models are in alphabetical order; author's last name is used if the model name is not available)

| Model | Aerenchynma | Diffusion | Ebullition | References |
|---|---|---|---|---|
| Arah's model | Yes | Yes | No | (Arah and Stephen, 1998) |
| Beckett's model | Yes | Yes | No | (Beckett et al., 2001) |
| "Cartoon" model | Yes | Yes | Yes | (Arah and Kirk, 2000) |
| CASA | Yes | Yes | Yes | (Potter, 1997; Potter et al., 1996) |
| CH4MOD | Yes | Yes | Yes | (Huang et al., 1998b; Huang et al., 2004; Li et al., 2012) |
| Christensen's model | No | No | No | (Christensen et al., 1996) |
| CLASS | No | Yes | No | (Curry, 2009; Curry, 2007) |
| CLM4Me | Yes | Yes | Yes | (Riley et al., 2011) |
| CLM-Microbe | Yes | Yes | Yes | (Xu et al., 2015; Xu et al., 2014) |
| DAYCENT | No | Yes | No | (Del Grosso et al., 2002; Del Grosso et al., 2009; Del Grosso et al., 2000) |
| Ding's model | Yes | No | No | (Ding and Wang, 1996) |
| DLEM | Yes | Yes | Yes | (Tian et al., 2010; Xu and Tian, 2012) |
| DNDC | Yes | Yes | Yes | (Li, 2000b) |
| *ecosys* | No | Yes | Yes | (Grant, 2001, 1998) |
| Gong's model | Yes | Yes | Yes | (Gong et al., 2013) |
| HH model | Yes | Yes | Yes | (Cresto-Aleina et al., 2015) |
| IAP-RAS | No | No | No | (Eliseev et al., 2008; Mokhov et al., 2007) |





| | | | | |
|---|---|---|---|---|
| Kettunen's model | Yes | Yes | Yes | (Kettunen, 2003) |
| Lovley's model | No | No | No | (Lovley and Klug, 1986) |
| LPJ-Bern | Yes | Yes | Yes | (Spahni et al., 2011) |
| LPJ-WHyMe | Yes | Yes | Yes | (Wania et al., 2010, 2009) |
| LPJ-WSL | No | No | No | (Hodson et al., 2011) |
| Martens' model | Yes | Yes | Yes | (Martens et al., 1998) |
| MEM | No | No | No | (Cao et al., 1995; Cao et al., 1998) |
| MERES | Yes | Yes | No | (Matthews et al., 2000) |
| Nouchi's model | Yes | Yes | No | (Hosono and Nouchi, 1997; Nouchi et al., 1994) |
| ORCHIDEE | Yes | Yes | Yes | (Ringeval et al., 2010; Ringeval et al., 2011) |
| Ridgwell's model | No | Yes | No | (Ridgwell et al., 1999) |
| SDGVM | No | No | No | (Hopcroft et al., 2011) |
| Segers' model | Yes | Yes | Yes | (Segers and Kengen, 1998; Segers and Leffelaar, 2001a, b; Segers et al., 2001) |
| Tagesson's model | No | No | No | (Tagesson et al., 2013) |
| TEM | Yes | Yes | Yes | (Zhuang et al., 2004) |
| TRIPLEX-GHG | Yes | Yes | Yes | (Zhu Q. et al., 2014) |
| UW-VIC | Yes | Yes | Yes | (Bohn and Lettenmaier, 2010; Bohn et al., 2007) |
| van Bodegom's model | Yes | Yes | Yes | (van Bodegom et al., 2000; Van Bodegom et al., 2001) |
| VISIT | Yes | Yes | Yes | (Inatomi et al., 2010; Ito and Inatomi, 2012) |
| De Visscher's | No | Yes | No | (De Visscher and Van |





| model | | | | Cleemput, 2003) |
|---|---|---|---|---|
| Walter's model | Yes | Yes | Yes | (Walter and Heimann, 2000; Walter et al., 1996) |
| Xu's model | Yes | Yes | Yes | (Xu et al., 2007) |

95



Table 2. Key mechanisms/features of CH$_4$ processes and their representations in CH$_4$ models

| Key mechanisms | Models |
|---|---|
| Methanogenesis | Arah's model, "Cartoon" model, CASA, CH4MOD, Christensen' model, CLM4Me, CLM-Microbe, Ding's model, DLEM, DNDC, *ecosys*, Gong's model, IAP-RAS, Kettunen's model, Lovley's model, LPJ-Brn, LPJ-WHyMe, LPJ-WSL, Martens' model, MEM, MERES, ORCHIDEE, SDGVM, Segers' model, TEM, TRIPLEX-GHG, UW-VIC, van Bodegom's model, VISIT, Walter's model, Xu's model |
| Methanotrophy | Arah's model, "Cartoon" model, CASA, CLASS, CLM4Me, CLM- Microbe, DAYCENT, DLEM, DNDC, *ecosys*, Gong's model, Kettunen's model, LPJ-Bern, LPJ-WHyMe, Martens' model, MEM, MERES, ORCHIDEE, Ridgwells model, SDGVM, Segers' model, TEM, TRIPLEX-GHG, UW-VIC, van Bodegom's model, VISIT, De Visscher's model, Wlater's model, Xu's model |
| Anaerobic oxidation of CH$_4$ | CLM-Microbe |
| Substrate (Acetate/DOC) | CH4MOD, CLM-Microbe, DLEM, DNDC, *ecosys*, Gong's model, Kettunen's model, Lovley's model, Martens'model, MEM, MERES, SDGVM, Segers' model, van Bodegom's model, Xu's model |
| Microbial functional groups | CLM-Microbe, DNDC, *ecosys* |
| CH$_4$ storage in soil profile | Arah's model, Beckett's model, "Cartoon" model, CLM4Me, CLM-Microbe, *ecosys*, Kettunen's model, Martens' model, MERES, Nouchi's model, ORCHIDEE, Segers' model, UW-VIC, van Bodegom's model, VISIT, De Visscher's model, Walter's model |
| O$_2$ availability for CH$_4$ oxidation | Arah's model, Beckett's model, "Cartoon" model, CLM4Me, CLM-Microbe, *ecosys*, Kettunen's model, MERES, Segers' model, van Bodegom's model, De Visscher's model, Xu's model |
| Iron biogeochemistry | van Bodegom's model |
| Sulfate biogeochemistry | Lovley's model, Martens' model, van Bodegom's model |
| Frozen trapped CH$_4$ | None |
| Embedded in Earth System Model | CLM4Me, CLM-Microbe, IAP-RAS, ORCHIDEE, SDGVM |





| Vertical resolved biogeochemistry | Arah's model, Beckett's model, "Cartoon" model, CLASS, CLM4Me, CLM-Microbe, DNDC, *ecosys*, Gong's model, HH model, IAP-RAS, Kettunen's model, Lovley's model, LPJ-Bern, LPJ-WHyMe, LPJ-WSL, Martens' model, MERES, ORCHIDEE, Ridgwell's model, SDGVM, Segers' model, TRIPLEX-GHG, UW-VIC, VISIT, De Visscher's model, Walter's model, Xu's model |
|---|---|
| Regional-scale, capacity for up-scaling | CASA, CH4MOD, Christensen's model, CLASS, CLM4Me, CLM-Microbe, DAYCENT, DLEM, *ecosys*, Gong's model, HH model, IAP-RAS, LPJ-Bern, LPJ-WHyMe, LPJ-WSL, Martens' model, MEM, MERES, ORCHIDEE, Ridgwell's model, SDGVM, Tagesson's model, TEM, TRIPLEX-GHG, UW-VIC, VISIT, Walter's model |




Table 3. The mathematical equations used to described the CH$_4$ processes used in representative models ($P_{CH4}$ is the CH$_4$ production rate; $Oxid_{CH4}$ is the CH$_4$ oxidation rate; $T_{CH4}$ is the CH$_4$ transport rate; $D_{CH4}$ is the CH$_4$ diffusion rate; some parameter may have been changed from original publication to keep relatively consistent in this table)

| CH$_4$ processes | | Equations | Ecological description | Model examples |
|---|---|---|---|---|
| CH$_4$ substrate and CH$_4$ production | 1 | $P_{CH_4} = f(T, W)$ | A function of temperature (T) and moisture (W) | Christensen's model, IAP-RAS, DAYCENT |
| | 2a | $P_{CH_4} = r \times HR \times f(T, W)$ | A portion of heterotrophic respiration, affected by temperature (T) and moisture (W) | LPJ family, CLM4Me, Ding's model, MERES, TRIPLEX-GHG |
| | 2b | $P_{CH_4} = r \times SOM \times f(T, W)$ | A portion of soil organic matter (SOM), affected by temperature (T) and moisture (W); Walter's model use indirect association with NPP | CH4MOD, Gong's model, HH model, Walter's model |
| | 3 | $P_{CH_4} = V \times \dfrac{[DOC]}{K_{DOC} + [DOC]} \times f(T, W)$ | A portion of dissolved organic carbon (DOC), affected by temperature (T) and moisture (W) | MEM, DLEM |
| | 4 | $P_{CH_4} = f(DOC, Acetate, CO_2) \times f(T, W)$ | Mechanistic processes for CH$_4$ production are considered, affected by temperature (T) and moisture (W) | Kettunen's model, Segers' model, van Bodegoms model, and *ecosys* |
| CH$_4$ oxidation | 5 | $Oxid_{CH_4} = V \times \left( \dfrac{[CH_4]}{K_{CH_4} + [CH_4]} \right) \times f(T, W)$ | Oxidation as a function of CH$_4$ concentration and temperature and moisture | DLEM, TRIPLEX-GHG, VISIT |





| | | | | |
|---|---|---|---|---|
| | 6 | $Oxid_{CH_4} = V \times \left(\dfrac{[CH_4]}{K_{CH_4} + [CH_4]}\right)\left(\dfrac{[O_2]}{K_{O_2} + [O_2]}\right) \times f(T,W)$ | Oxidation as a function of $CH_4$ and $O_2$ concentration, temperature and moisture | Arah's model, Cartoon model, CLM4Me, CLM-Microbe, Kettunen's model |
| $CH_4$ transport | 7 | $T_{CH_4} = V * ([CH_4] - \overline{[CH_4]})$ | $V$ is the parameter for distance, diffusion coefficient, etc.; $[CH_4]$ is the concentration of $CH_4$ in the soil/water profile (dissolvability for DLEM, 0 for DNDC); and $\overline{[CH_4]}$ is the threshold of $CH_4$ concentration above which $CH_4$ will be transported to the atmosphere via either of the three transport pathways | DLEM, DNDC, Walter's model |
| | 8a | $A = \dfrac{C(z) - C_a}{r_L z / D + r_a} \, pT\rho_r$ | *Aerenchyma transport* | CLM4Me |
| | 8b | Moves to first unsaturated layer and then released to gaseous phase | *Ebullition* | CLM4Me |
| | 8c | $D_{CH_4} = D \times \dfrac{\Delta[CH_4]}{\Delta z}$ | Diffusion of $CH_4$ was simulated following Fick's law; CLM4Me separate aqueous and gaseous diffusion | CLM4Me, CLM-Microbe, *ecosys*, Ridgwell's model, TRIPLEX-GHG; Sergers' model |
| Temperature effects | 9 | $f(T) = a \times T + b$ $f(T) = a \times T^2 + b \times T + c$ $f(T) = b \times e^{0.2424 \times T}$ | Linear regression on temperature or degree days; DNDC simulate temperature impact on production not on oxidation | DAYCENT, DNDC, IAP-RAS, LPJ family |



| | 10 | $f(T) = Q_{10}^{\frac{(T-T_{ref})}{10}}$ | $Q_{10}$ equations; $T_{ref}$ is the reference temperature | CH4MOD, CLM-Microbe, CLM4Me, DLEM, VISIT, Kettunen's model |
|---|---|---|---|---|
| | 11a | $V_T = V^0 \times \exp\left(\frac{\Delta E}{R}\left[\frac{1}{T^0} - \frac{1}{T}\right]\right)$ | Arrhenius equation | Arah's model, Ding's model |
| | 11b | $f_T = \dfrac{T_s \times \exp\left(A - \frac{H_a}{R \times T_s}\right)}{\left[1 + \exp\left(\frac{H_{dl} - S \times T_s}{R \times T_s}\right) + \exp\left(\frac{S \times}{}\right.\right.}$ | Modified Arrhenius equation; $T_s$ is soil temperature at $K$; $A$ is the parameter for $f_T = 1.0$ at $T_s = 303.16$ K; $H_a$ is the energy of activation (J mol$^{-1}$); $R$ is universal gas constant (J mol$^{-1}$ K$^{-1}$); $H_{dl}$ and $H_{dh}$ are energy of low and high temperature deactivation (J mol$^{-1}$) | *ecosys* |
| Moisture effects on methanogenesis and methanotrophy | 12 | No moisture effect is simulated, rather inundation area is simulated | No equation, while a temporal and spatial variation of inundation and saturation impacts | CASA |
| | 13 | $F_\vartheta = e^{-P/P_c}$ | Water stress for oxidation, where P is soil moisture and Pe= -2.4×10$^5$ mm | CLM4Me |
| | 14 | $f(SM) = \begin{cases} 1, \\ \left[1 - \dfrac{log_{10}\varphi - log_{10}(0.2)}{log_{10}(100) - log_{10}(0.2)}\right]^\beta \\ 0, \end{cases}$ | β is an arbitrary constant, $^\phi$ is the soil water potential | CLASS |



| | 15 | $f_{prod}(SM) = \left(\dfrac{SM - SM_{fc}}{SM_{sat} - SM_{fc}}\right)^2$ $\times 0.368$ $\times e^{\left(\frac{SM - SM_{fc}}{SM_{sat} - SM_{fc}}\right)}$ $f_{oxid}(SM) = 1 - f_{prod}(SM)$ | Different impacts on CH$_4$ production and consumption; SM: soil moisture; $SM_{fc}$: field capacity; $SM_{sat}$: saturation soil moisture | DLEM |
|---|---|---|---|---|
| | 16 | $f(SM)$ $= \dfrac{(M_V - M_{min}) \times (M - M_n}{(M_V - M_{min}) \times (M_V - M_{max}) - ($ | Bell-shape curve | TEM |
| pH effects | 17a | $f(pH)$ $= \dfrac{(pH - pH_{min}) \times (pH - p}{(pH - pH_{min}) \times (pH - pH_{max}) -}$ | Bell-shape curve | CLM-Microbe, MEM, TEM, |
| | 17b | $f(pH)$ $= 10^{-0.2335 \times pH^2 + 2.7727 \times pH - 8.6}$ | Bell-shape curve | CLM4Me |
| | 17c | $f(pH)$ $= \begin{cases} 0 & pH \le 4 \\ \dfrac{1.02}{1 + 1000000 \times e^{(-2.5 \times pH)}} \\ \dfrac{1.02}{1 + 1000000 \times e^{(-2.5 \times (14 - pH))}} \end{cases}$ | Bell-shape curve | DLEM |





Table 4. Temperature dependence of $CH_4$ processes in various models (blank indicates the $Q_{10}$ function is not used; all temperatures are expressed as °C, 273.15 was used for unit conversion)

| Model | $Q_{10}$ | Reference temperature (°C) | Note | Sources |
|---|---|---|---|---|
| CASA | | | Based on a linear equation with temperature | (Potter, 1997) |
| DAYCENT | | | Linear equation y = 0.209 * T + 0.845 | (Del Grosso et al., 2000) |
| LPJ family LPJ-Bern LPJ-WHyMe LPJ-WSL | | | Linear function was used for temperature impacts on diffusion | (Hodson et al., 2011; Spahni et al., 2011; Wania, 2007) |
| Christensen's model | 2 | 2 | For temperature > 0, the temperature impact is set to zero when < 0 | (Christensen and Cox, 1995) |
| CH4MOD | 3 | 30 | T=30 for 30 < T ≤ 40 | (Huang et al., 1998b) |
| CLM4Me | 2 | 2 | Parameters for baseline simulation | (Riley et al., 2011) |
| CLM-Microbe | 1.5 | 13.5 | | (Xu et al., 2015) |
| DLEM | 2.5 | 30 | For a temperature range of [-5, 30]; temperature impact is set to zero when < -5 or > 30 | (Tian et al., 2010) |
| Kettunenn's model | 4.0 for production, 2.0 for oxidation | 10 | Standard $Q_{10}$ function | (Kettunen, 2003) |
| ORCHIDEE | Abisko site, 2.6; Michigan site, 3.2; Panama site, 1.2 | Mean annual temperature | $Q_{10}$ function with different parameters across biomes | (Ringeval et al., 2010) |





| TEM | Alpine tundra: wetland, 3.5; upland, 0.8. Wet tundra: wetland, 2.2; upland, 1.1. Boreal forest: wetland, 1.9; upland, 1.5 | Alpine tundra: wetland, -3.0; upland, 8.0. Wet tundra: wetland, -5.5; upland, 8.0. Boreal forest: wetland, 1.0; upland, 7.0 | $Q_{10}$ function with different parameters across biomes | (Zhuang et al., 2004) |
|---|---|---|---|---|
| TRIPLEX-GHG | 1.7-16 for production, 1.4-2.4 for oxidation | 25 for optimal, 45 for highest temperature | Modified $Q_{10}$ equation | (Zhu et al., 2014a) |
| VISIT | | Mean annual temperature | | (Ito and Inatomi, 2012) |
| Walter's model | 2 | Ombrotrophic bog, 12; poor fen, 6.5; oligotrophic pine fen, 3.5; Arctic tundra, 0; swamp, 27 | $Q_{10}$ function with different parameters across biomes | (Walter and Heimann, 2000) |
| Arah's model | | 10 | Arrhenius equation | (Arah and Stephen, 1998) |
| *ecosys* | | 30 | Modified Arrhenius equation | (Grant et al., 1993) |

05





**Figure legend**

Figure 1. The published $CH_4$ models and modeling trends in terms of applicability and mechanistic representation of $CH_4$ cycling processes at decadal-scale and the envisioned $CH_4$ model capability

Figure 2. Percentage of $CH_4$ models with consideration of some key $CH_4$ mechanisms. The percentage was calculated as the number of models considering each mechanisms divided by the total number of published models in each time period.

Figure 3. Three types of models with key mechanisms for $CH_4$ production and oxidation (*SOM*: Soil organic matter; *NPP*: net primary production; *DOC*: dissolved organic carbon; $O_{atm}$: oxidation of atmospheric $CH_4$; *P*: plant-mediated transport; *D*: diffusion transport; *E*: ebullition transport; $O_{xid}$: oxidation; $O_{trans}$: oxidation of $CH_4$ during transport)

Figure 4. Key features of future mechanistic $CH_4$ models with a full representation of primary $CH_4$ processes in the terrestrial ecosystems. The data assimilation system and model benchmarking system are also shown as auxiliary components to the future $CH_4$ models.





Cartoon in 2000; CLASS in 2000; DAYCENT in 2000; DNDC in 2000; MERES in 2000; Beckett in 2011; De Visscher in 2003; IAP-RAS in 2007; TEM in 2004; Kettunen in 2003; LPJ-WHyMe in 2009; ORCHIDEE in 2008; PEATLAND-VU in 2006; WU-VIC in 2007; van Bogedom in 2001; LPJ-WHyMe in 2007; Xu in 2007

CLM-Microbe in 2015; DLEM in 2010; CLM4Me in 2011; Gong's model in 2013; HH model in 2015; Tagesson in 2013; TRIPLEX-GHG in 2014; ORCHIDEE-2010; VISIT in 2010; LPJ-Bern in 2011; LPJ-WSL in 2011

Nouchi in 1994; MEM in 1995; Christensen, 1996; Ding in 1996; Walter's model in 1996; CASA in 1996, 1997; Arah's model in 1998; *ecosys* in 1998; Martens' model in 1998; CH4MOD in 1998; Segers in 1998; Ridgwell in 1999;

Lovely's model in 1988

**1980s**

**1990s**

**2000s**

**2010s**

Mechanistic models for understanding $CH_4$ processes

Mechanistic models for understanding $CH_4$ processes; plot- and regional simulations for quantifying $CH_4$ budget

Plot-level model development for $CH_4$ cycling; and regional model for quantifying $CH_4$ budget; mechanistic models for understanding $CH_4$ processes;

Regional model for quantifying $CH_4$ budget; mechanistic models for understanding $CH_4$ processes; plot-level model development for $CH_4$ cycling

Integrative models capable to fuse multiple sources data; mechanistic model with primary $CH_4$ cycling including production, oxidation, transport, and environmental controls

**Theoretical analysis; mechanistic understanding**

**Applicable on budget estimation at plot- and regional-scales; integration tool**

Fig. 1.



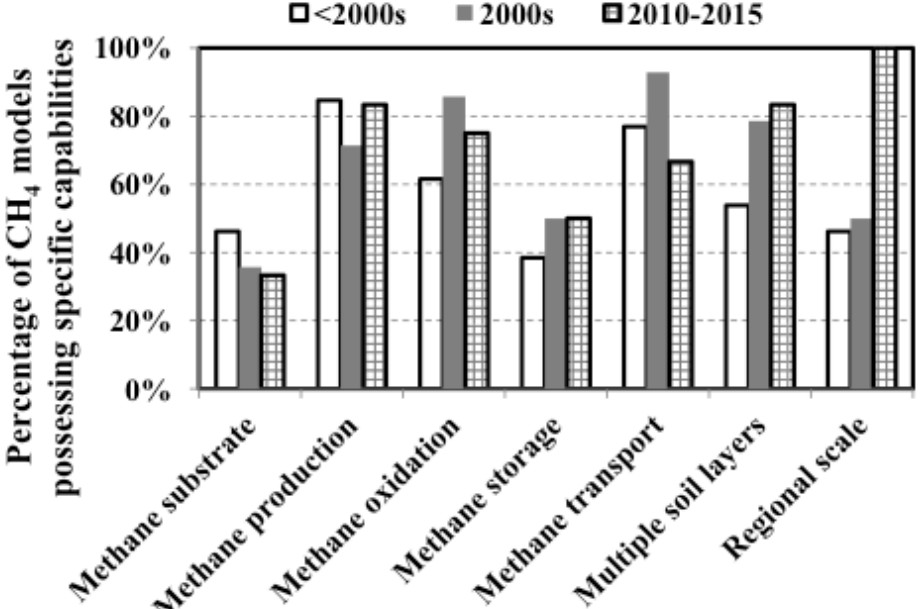

Fig. 2.



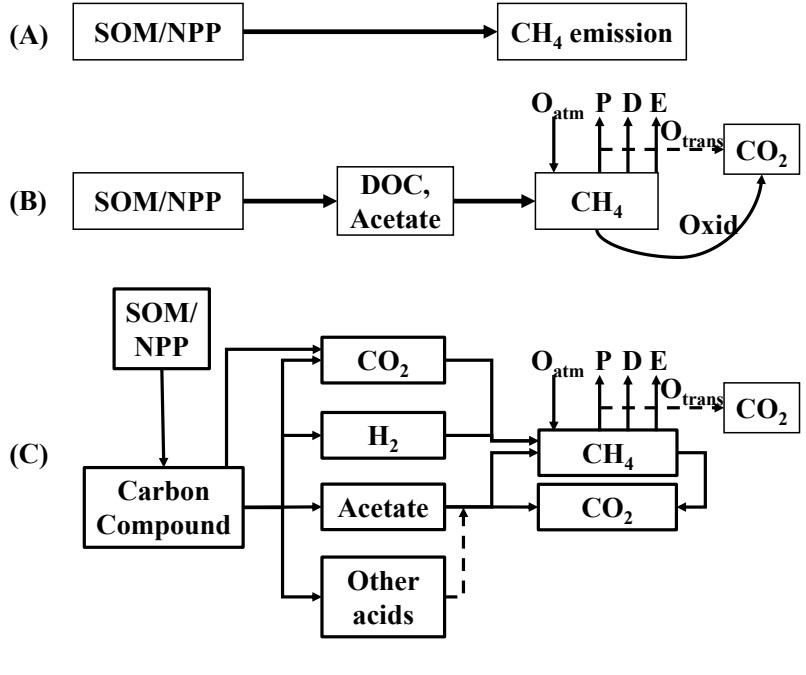

Fig. 3.



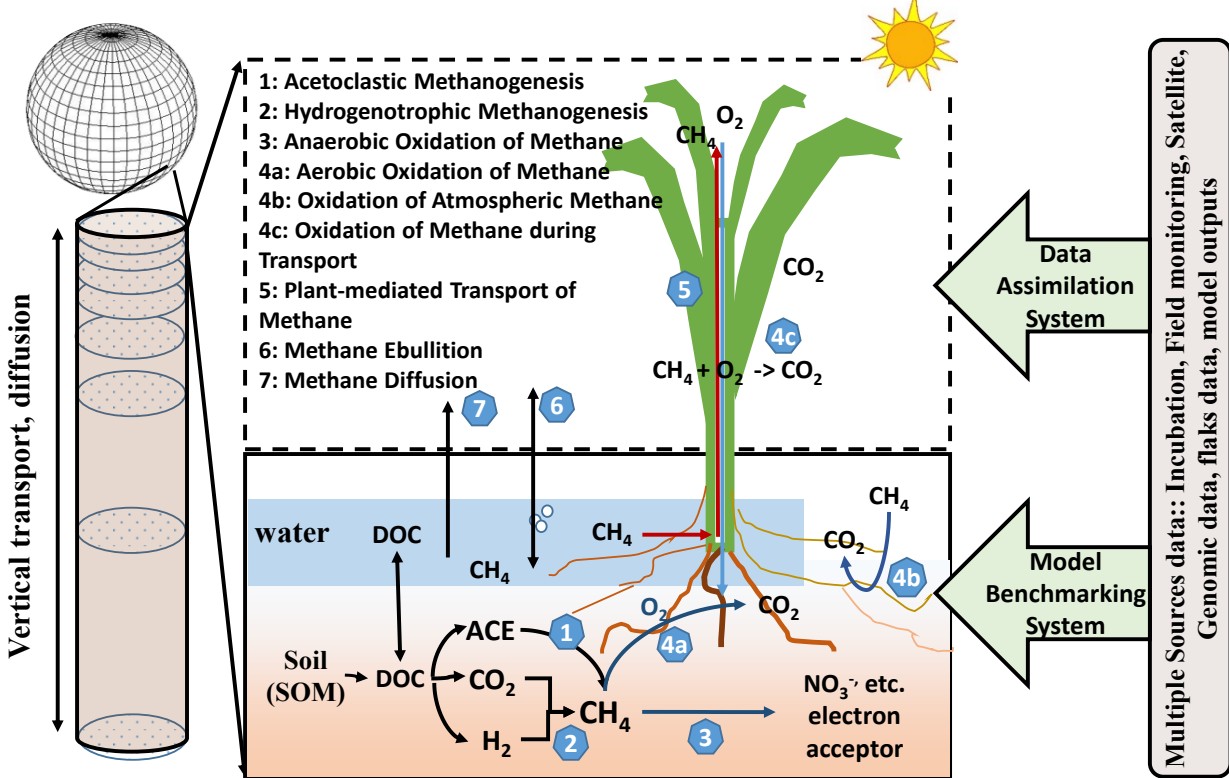

Fig. 4