# Peer review of "Reviews and syntheses: Four Decades of Modeling Methane Cycling in Terrestrial Ecosystems"

_Biogeosciences, 2016_

## Referee Comment (RC1) · Anonymous Referee #1 · 16 Mar 2016

Review of "Reviews and syntheses: Four Decades of Modeling Methane Cycling in Terrestrial Ecosystems" by Xiaofeng Xu et al. submitted to Biogeosciences Discussion.

General comments: This manuscript provides an overview and a synthesis of the evolution of models focusing on methane emissions from terrestrial ecosystems. The manuscript is based on a comparison among 39 methane models described in peer-reviewed articles, followed by a general synthesis that includes outlines of future challenges and directions in the field. I read the review with interest; it is a review which as far as I know has not been done before. Understanding the current state and potential future challenges of methane modelling will be of interest both to field researchers and for new modelling projects. The manuscript also has shortcomings that I think should be addressed before publication is considered. First, I find that the overall presentation can be improved for increased clarity, particularly with regards to sentence and

paragraph structure. I have several examples in the specific comments below, but an overall assessment is recommended.

I also think the introduction could do a better job in outlining the scope of the manuscript, particularly I would favor some more specific information, e.g "first we will give an overview of the range of processes that have been considered in methane models, based on this we will classify existing models as determined by the range of processes considered. The following sections will review and synthesize how models deal specifically with methane production, consumption and transport within soils. . . . etc." I also recommend the authors to better define several key concepts in the manuscript. This would include your definition of a terrestrial ecosystem (see further comment below), and a definition of what constitutes a "primary process" with regards to methane dynamics (is this just your ranking of which processes that are more likely to have a stronger influence on the resulting emissions magnitude?).

The term "terrestrial ecosystems" is particularly important for this manuscript, since it defines the scope of the models that have been reviewed. How do you define terrestrial ecosystems? I.e. what is the distinction from aquatic ecosystems and why are not models of aquatic ecosystem methane emissions considered in this review? The review emphasizes the need to be able to estimate methane emissions at large regional to global scales, but aquatic ecosystems might (depending on your definition) have greater emissions than terrestrial ecosystems at the regional/global scale, so the omission of aquatic ecosystems is important. How do you define wetlands in terms of being terrestrial or aquatic ecosystems? The US and Canadian definitions of wetlands include open water wetlands with up to 2 m of standing water – are all these considered terrestrial in this review? Would it be considered a future challenge to extent the current models to include aquatic ecosystems, particularly streams, rivers, ponds and lakes?

Another topic that I do not think get sufficient attention in the manuscript relate to the diversity of goals for different models, and how that influences the choices made in the

model development. In the introduction you bring up the fact that models can be developed for extrapolation to regional or global scales, or for process-level models that are developed to understand methane dynamics at the site level. The latter type of model requires information on many site-specific parameters (soil microbial community, iron and sulfur data etc etc), data which is not available for large regions. One recommendation in this manuscript it that more processes should be considered for methane model – however, for models aimed at regional to global scales this is likely to lead to highly unconstrained models since the data to run the models does not exist and is highly unlikely to be mapped. In short, I think there is a need to discuss how modelling goals will influence model development, particularly how this relates to available model data inputs.

The issue of spatial data availability, used as model input, is also not discussed in the manuscript. It is my belief is that improved spatial data on wetland extents and wetland characteristics are likely to improve our accuracy of regional to global estimates of methane emissions (both magnitude and spatial patterns) much more than the incorporation of additional processes in the models. The use of different spatial products (wetland maps, inundation maps etc) for estimating global methane emissions is known to produce wildly different spatial patterns of regional methane emissions. I believe a discussion on how available data, and the use of available data, affect model development and modelling results deserve some attention in this review.

Specific comments:

P2 L37. I strongly discourage use the concept of global warming potential when discussing methane emissions from wetlands. GWP are only applicable when considering "new" sources, i.e. changes in emissions, but cannot be used when evaluating sustained emissions. Wetlands have been emitting methane for millennia, thus their methane emissions have a much lesser additional impact on climate forcing at this point than would be concluded based on GWP (unless they have increased as a result of climate change or by other means). See Frolking and Roulet et al 2007 Glob Change

Biol.

P3 L70-72. This is a weak sentence to finish an introduction.

P3 L74-76. Is it possible to reference the original sources?

P4 L83. What is meant by "primary CH4 processes"? How do you distinguish primary processes from other processes? Do these primary processes include the 3 methanogenesis processes, 2 methanotrophy processes, and the 7 transport mechanisms? Several of these processes, which I assume are what you consider primary processes since they are listed in the sentence after your statement on primary processes, are not discussed in regards to how they are represented in models. E.g. methylotrophic methanogenesis is only mentioned once, and is not discussed with regards to how it is considered by models. Also, of the seven transport mechanisms you only discuss ebullition, diffusion and plant-mediated transport – what are the other four processes? Overall, I think you need a better framework for how you classify the different processes, including a motivation on why some of these processes are to be considered in the review and why other are not.

P4 L85. Clumsy sentence structure, omit "(depending on how one counts)".

P4 L87. Importance in time and space – and you should probably highlight that it varies by wetland characteristics.

P4 L92. Perhaps a brief description is needed that explains the differences between acetoclastic and hydrogenotrophic processes, in terms of under what conditions they are more likely to dominate and why.

P4 L107. This is a awkward way of saying that upland soils are net sinks of atmospheric methane.

P4-5 L09-15. This sentence is very long and introduced several new concepts not previously described.

P5 L16. This is the third time I have seen the same point being raised already - "process vary significantly depending on temporal and spatial scales".

P5 L17. How do you define direct and indirect effects with regards to wetland methane emissions? It is not clear to me given the examples brought up. Is the classification of direct and indirect processes different from that of primary and other processes introduced earlier?

P5 L35. "Water sediments", do you mean "Aquatic sediments"?

P5-P6 L36-57. I'm not sure this listing of the different methane models is effective. I would recommend merging this section with the section below (L181-199) on the different groups of model, i.e. to bring these models up as examples of each group.

P7 L66. What is your definition of regional simulation capability? This has not been presented.

P8 L09. Do you have any field data that can support your statement that substrate characterization is key for modelling methane production?

P10 L62. Unclear if you mean the third group of the three groups described in the "Model Classification' section or the third group described in the "Methanogenesis" section. I would recommend separating the models into groups once, rather than a new division of models in each section.

P10 L264. Seems appropriate to discuss substrate limitation and Michelis-Menten dynamics of methanogenesis in the section on methanogenesis rather than methanotrophy.

P17, L43. This sentence has poor structure, also, what is meant by "was not included in any of the three groups because that effort will likely be achieve over the long term"?

P19, L09. Can you give examples of less-studies ecosystems?

P20, L31. Sentence structure: "integration between model development and data collection is much stronger for advancing science", do you mean that integration is important for advancing our scientific understanding of methane dynamics?

---

## Referee Comment (RC2) · Anonymous Referee #2 · 18 Apr 2016

Comments on "Reviews and synthesis: Four decades of modeling methane cycling in terrestrial ecosystems" by Xiaofeng Xu et al. submitted to Biogeosciences.

General comments

In this manuscript, the authors reviewed 39 terrestrial methane models and discussed their limitations and future opportunities. This kind of model review has been partly conducted in introduction of model intercomparison project (e.g., WETCHIMP; Melton et al., 2013, Wania et al., 2013), but I agree that this manuscript gives a more thorough overview. The 39 models were classified into several categories (or generations) from the points of processes and complexity. Also, the authors gave good overview of underlying mechanisms of methane production, consumption, and transportation. In the light of its importance as the second important anthropogenic greenhouse gas, this

manuscript is timely and within the scope of the journal.

The manuscript is fairly prepared, but I have several recommendations. First, I felt redundancies in the manuscript. For example, influential factors of methane processes are similarly listed in Page 5 Line 118 and Page 12 Line 322. I recommend refining the manuscript by reducing redundancies. Second, I recommend giving a broader picture of terrestrial models that include methane processes. The authors mentioned that methane schemes would be implemented into Earth system models (ESMs). Similarly, integrated terrestrial models (other than ESMs) should include methane processes to evaluate e.g. the effect of mitigation practices. Overall, I recommend that the manuscript be worth publication after moderate to major revision.

Specific comments

Page 3 Line 65

This manuscript does not cover several quantitatively important processes such as methane emissions from biomass burning, termites, and ruminants. Please justify here for ignorance of these processes.

Page 5 Line 133

In the 1980s, E. Mattews and I. Fung (1987) achieved a pioneering work in which not only terrestrial but also atmospheric methane dynamics were simulated at the global scale. I think that their work should be mentioned in text.

Page 6 Line 159

In Figure 6 of Wania et al. (2013), estimations of methane production area in the contemporary models are well summarized.

Page 7 Line 190

Can you give several examples for the second group model?

[Figure]

Page 8 Line 193

Can you give several examples for the third group model?

Page 9 Line 233

Can you show the 31 models by adding a column in Table 1?

Page 9 Line 244

"address" should be "addressed".

Page 10 Line 246 and Table 1

In addition to Ridgwell et al. (1999), several methane oxidation models have been presented and could be mentioned here: e.g., Del Grosso et al. (2000) and Curry (2007).

Page 10 Line 251

Can you indicate a typical value of the contribution of anaerobic methane oxidation in total oxidation?

Page 11 Line 275

In terms of the modeling of vertical profile, parameterization of methane diffusion coefficient within soil is critically important. Do you agree?

Page 13 Line 35

Yvon-Durocher et al. (2014) implied that the temperature response of methane emission would be evaluated using a single consistent model. If correct, the divergence in present models would be largely reduced. Do you agree?

Page 14 Line 356

As long as I know, only a few global dataset of soil pH is available. Also, in situ measurement and model prediction of soil pH are rather difficult. I think these difficulties in

using soil pH should be noted.

Page 15 Line 380

It looks wired to give a summary at this place, because it is usually given at the end of the manuscript. Actually, the statements around Page 16 Line 411 are as if your conclusion.

Page 18 Line 460

A few more processes not mentioned here have been presented: e.g., emission from tank bromeliads (Martinson et al., 2010) and emission from small ponds (Holgerson and Raymond, 2016).

Page 19 Line 504

I recommend adding one more (6th?) challenge. Modeling of human-natural processes such as emission from managed ponds and estuaries is important in terms of mitigation. Namely, we should consider both natural biogeochemical processes and human management effects.

Page 21 Line 540

Do you mean "Markov Chain Monte Carlo (MCMC)"?

Page 25 Line 623

Please correct information for Bohn et al. (2015):

Bohn, T. J., Melton, J. R., Ito, A., Kleinen, T., Spahni, R., Stocker, B. D., Zhang, B., Zhu, X., Schroeder, R., Glagorev, M. V., Maksyutov, S., Brovkin, V., Chen, G., Denisov, S. N., Eliseev, A. V., Gallego-Sala, A., McDonald, K. C., Rawlins, M. A., Riley, W. J., Subin, Z. M., Tian, H., Zhuang, Q., and Kaplan, J. O.: WETCHIMP-WSL: Intercomparison of wetland methane emissions over West Siberia, Biogeosciences, 12, 3321–3349, doi: 10.5194/bg-12-3321-2015, 2015.

Figure 4

Can you include the microbial community factor into the figure?

References

Curry, C. L.: Modeling the soil consumption of atmospheric methane at the global scale, Global Biogeochem. Cycles, 21, doi:10.1029/2006GB002818, 2007.

Del Grosso, S. J., Parton, W. J., Mosier, A. R., Ojima, D. S., Potter, C. S., Borken, W., Brumme, R., Butterbach-Bahl, K., Crill, P. M., Dobbie, K., and Smith, K. A.: General CH4 oxidation model and comparisons of CH4 oxidation in natural and managed systems, Global Biogeochem. Cycles, 14, 999-1019, 2000.

Holgerson, M. A., and Raymond, P. A.: Large contribution to inland water CO2 and CH4 emissions from very small ponds, Nature Geoscience, 9, 222–226, doi:10.1038/NGEO2654, 2016.

Martinson, G. O., Werner, F. A., Scherber, C., Conrad, R., Corre, M. D., Flessa, H., Wolf, K., Klose, M., Gradstein, S. R., and Veldkamp, E.: Methane emission from tank bromeliads in neotropical forests, Nature Geoscience, 3, 766–769, doi:10.1038/ngeo980, 2010.

Matthews, E., and Fung, I.: Methane emission from natural wetlands: global distribution, area, and environmental characteristics of sources, Global Biogeochem. Cycles, 1, 61-86, 1987.

Melton, J. R., Wania, R., Hadson, E. L., Poulter, B., Ringeval, B., Spahni, R., Bohn, T., Avis, C. A., Beerling, D. J., Chen, G., Eliseev, A. V., Denisov, S. N., Hopcroft, P. O., Lettenmaier, D. P., Riley, W. J., Singarayer, J. S., Subin, Z. M., Tian, H., Zürcher, S., Brovkin, V., van Bodegom, P. M., Kleinen, T., Yu, Z. C., and Kaplan, J. O.: Present state of global wetland extent and wetland methane modelling: conclusions from a model inter-comparison project (WETCHIMP), Biogeosciences, 10, 753–788, doi:10.5194/bg-10-753-2013, 2013.

Wania, R., Melton, J. R., Hodson, E. L., Poulter, B., Ringeval, B., Spahni, R., Avis, C. A., Chen, G., Eliseev, A. V., Hopcroft, P. O., Riley, W. J., Subin, Z. M., Tian, H., van Bodegom, P. M., Kleinen, T., Yu, Z. C., Singarayer, J. S., Zürcher, S., Lettenmaier, D. P., Beerling, D. J., Denisov, S. N., Prigent, C., Papa, F., and Kaplan, J. O.: Present state of global wetland extent and wetland methane modelling: methodology of a model inter-comparison project (WETCHIMP), Geoscientific Model Development, 6, 617–641, 10.5194/gmd-6-617-2013, 2013.

---

## Referee Comment (RC3) · Anonymous Referee #3 · 19 Apr 2016

Comments on the manuscript by Xu et al "Reviews and syntheses: Four decades of modeling methane cycling in terrestrial ecosystems" submitted to Biogeosciences Discussions.

Overall Evaluation

This manuscript presents a review of approaches used to model methane dynamics in terrestrial ecosystems in the last four decades. The review largely focuses on describing the variability in structure and mathematical descriptions of processes among 39 terrestrial methane models. Parameterization issues are touched upon in the section on environmental controls, mostly with respect to variability in Q10 (which affects temperature sensitivity of processes). The discussion makes suggestions for adding more complexity to methane models, primarily along the lines of more explicitly considering

microbial processes and dynamics. The discussion finishes with identifying knowledge gaps, modeling challenges, data needs, and the need for data-model integration.

This manuscript tries to cover a lot of ground. The primary strength of the manuscript, in my opinion, is largely in the description of variability in mathematical descriptions of processes. The other aspects of the review didn't provide a lot insight in my opinion, as the issues discussed were in many cases just touched upon and were not well developed. My main concern about this manuscript is that in trying to cover a lot of ground, it covers some of that ground poorly. I think there are several issues to address to improve the review. First, I think there are some general organization issues that could be addressed to improve the manuscript. Second, there are a number of cases in the presentation of putting the "cart" before the "horse". Third, I didn't find that the description in the variability in structure (as depicted in Figure 3) was based on an objective evaluation of the 39 terrestrial models. Fourth, there a number of assertions in the manuscript that should be presented as more open issues. Fifth, the challenge of scaling is only touched upon in the manuscript and needs to be better developed, and there is a need for some discussion of reconciliation with atmospheric data analyses. Sixth, beside the scaling/reconciliation issue, I also found several issues that need to be better developed/discussed including the modeling of ebullition, vertical representation of processes, model benchmarking, and data-model integration. Below I go into more depth on each of these issues, and finish my review with a listing of specific comments.

Issue 1: Organizational issues in the manuscript. The manuscript starts out well, but then gradually gets more and more disorganized. There is a lot of overlap of material between some of the later sections of the manuscript that could be eliminated with a more effective organization. Perhaps consider the organization of Luo et al. (2016, Global Biogeochemical Cycles), which review soil carbon models. The organization of that paper is (1) model structure, (2) model parameterization, and (3) external forcing. I think additional in this manuscript concerns scaling and reconciliation with atmospheric data. The strength of this manuscript is that it generally does a good job of reviewing

model structure, but a rather inadequate job of reviewing model parameterization, external forcing, scaling, and reconciliation issues.

Issue 2: "Cart" before the "Horse" issues. There are a number of places in the manuscript where the "cart" comes before the "horse", from the perspective of this being a review paper. For example, the citation to Figure 2 on line 162 talks about the timeline for inclusion of "key mechanisms", but these mechanisms haven't been described in a general sense yet. Table 2, which contains the list of "key mechanisms" isn't cited until line 175. Even when Table 2 is cited, the general reader gets no background on these mechanisms/features of models, as it is not used beyond a simple citation at the end of a sentence. Other rough spots in the manuscript involve adequately describing terms used in the manuscript. For example, acetoclastic and hydrogenotrophic methanogenesis suddenly appears on lines 238-240 without any prior description. "Advective transport" (line 203) is also not described.

Issue 3: Analysis of the variability in structure. What is the basis for defining three different types of models? It seems to me that this could be done in a much more objective fashion by doing some sort of cluster analysis among the 39 models reviewed in this study. Information from Tables 1, 2, and 3 could be put into an objective cluster analysis so that we better understood what factors seem to cause models to be distinct (or not distinct) from each other.

Issue 4: There are a number of assertions in the manuscript that have not been justified by any sort of rational analysis/argument. For example, why make a recommendation in the last sentence of section 4 (lines 196-198) on the third types of models as the means of moving forward with respect to improving reduced form models for application in Earth System Model applications? First of all, this is too early in the manuscript. Second, doesn't making this recommendation conflict with the sentence on lines 211-212 that the optimum complexity remains to be determined? At the end of section 6 there are four recommendations for models "based on the above-mentioned needs" and a citation to Figure 4. I didn't find the previous text in section 6 as being very

helpful for establishing these as the top needs. This all comes before the section 7, which talks about knowledge gaps and data needs. The arrows for benchmarking and data assimilation in Figure 4 have not been developed, and the issues of vertical transport/diffusion have only been touched upon. Also, the top recommendation that "the models (features?) should be embedded in an Earth System Model" seems strange to make here. The point here is that arguments have not been well enough organized and crafted to effectively make these recommendations. This sort of all gets back to issues 1 and 2 above. Finally, I can't say that I'm very fond of Figure 4 as being the synthetic figure for this manuscript – we've seen a lot of these sort of figures over the years. I suggest thinking about something that is truly synthetic based on this manuscript.

Issue 5: The issues of scaling and reconciliation with atmospheric data. Scaling is an important issue. It does pop up several places in the manuscript as a sort of "between the lines" issue, but it really needs its own section. I also think that the issue of reconciling model applications at particular scales with data from atmospheric analyses needs to be part of the discussion.

Issue 6: Other issues. I also found several issues that need to be better developed/discussed including the modeling of ebullition, vertical representation of processes, model benchmarking, and data-model integration. For example, transport mechanisms don't even show up as key features in Table 2, although they do appear somewhat in Table 1. These issues are touched upon in several places in the manuscript, but are not really effectively dealt with in a meaningful way.

Specific comments

Line 104-105: "contributes" is not really the right verb to use here. Just says "varies from 1 to 90%", for example.

Line 106-107: I really don't know what you mean by "oxidation of atmospheric CH4 contributes". Aren't all of the previous mechanisms in this paragraph ultimately oxidation of atmospheric CH4, albeit in the open pore space of the soil.

Line 109: Perhaps start a new paragraph after "methanotrophy.".

Lines 109-116: There is no information for the uninitiated reader to understand how these pathways differ from each other.

Line 120: I think this might be the only occurrence of "wind speed" in the manuscript. What do you mean by "wind speed" as an environmental factor.

Line 121: Define what you mean by "indirect" vs. "direct" environmental factors.

Line 147: I don't think Fan et al. (2013, Peatland DOS-TEM) has anything to do with the Zhuang et al. (2014) model in that it has a number of different features and to my understanding the two models do not share any code base.

Line 162: As mentioned earlier, the reader needs to know more about the key mechanisms before you present/interpret Figure 2.

Line 175: Need to make better use of Table 2 in the manuscript. As I indicated earlier, transport mechanisms need to be included in Table 2.

Line 213: Does use of "first group of models" refer to model types in Figure 3, or to the first set of empirical models referred to in the first paragraph of section 4.1?

Line 238-240: Where does the information on acetoclastic and hydrogenotropic methanogenesis appear in Table 3? Note that these production processes have not been defined for the reader.

Line 280: Why is Zhuang (2004) cited here in the context of immediately transporting CH4? This model is primarily a monthly model with a pseudo-daily time step. This transport issue is an important temporal scaling issue, and one which should appear in a separate section on temporal scaling.

Line 286: I think you should change "will likely" to "can".

Line 287: I think you should change "impossible" to "not straight forward".

[Figure]

Line 291: I note that ebullition is not adequately treated in this section (section 4.4).

Line 292: Why is this the "final" bottleneck, or why is even referred to as a "bottleneck".

Line 303: Define advective transport.

Line 313: I think you should change "most" to "some". Note that ebullition seems to be ignored in these three "transport" challenges. It is a dominant pathway in some systems.

Line 319: I note that the simulation of variability in some environmental controls is not adequately treated in section 4.5 on environmental controls.

Lines 331-332: I think that this sentence needs to refer to Eq 9, 10, and 11 instead of 10, 11, and 12. Note that the third function in Eq 9 is essentially equivalent to Eq 10 in that the Q10 can be derived from the exponent.

Line 347: I think you mean Eqs. 13-16 instead of 12-15.

Lines 356-367: Do any models represent pH variability in time? It would be useful to know how models represent pH variability in space.

Lines 393-394: Why is the comparison of high frequency observational data needed for future model-model inter-comparison? I think it would be most important to high quality seasonal and interannual estimates derived from observations to effectively test and compare models.

Line 405: With respect to shifts, are you referring to shifts in time or in space?

Line 479: What do you mean by "order 1-10". Do you mean by a "factor of 1-10"? The language could be confused for "orders of magnitude".

Luo, Y., A. Ahlstrom, S.D. Allison, N.H. Batjes, V. Brovkin, N. Carvalhais, A. Chappell, P. Ciais, E.A. Davidson, A. Finzi, K. Georgiou, B. Guenet, O. Hararuk, J.W. Harden, Y. He, F. Hopkins, L. Jiang, C. Koven, R.B. Jackson, C.D. Jones, M.J. Lara, J. Liang,

A.D. McGuire, W. Parton, C. Peng, J.T. Randerson, A. Salazar, C.A. Sierra, M.J. Smith, H. Tian, K.E.O. Todd-Brown, M. Torn, K.J. van Groenigen, Y.P. Wang, T.O. West, Y. Wei, W.R. Wieder, J. Xia, X. Xu, X. Xu, and T. Zhou. 2016. Toward more realistic projections of soil carbon dynamics by Earth system models. Global Biogeochemical Cycles 30:40-56, doi:10.1002/2015GB005239.
* * *

---

## Referee Comment (RC4) · Anonymous Referee #4 · 19 Apr 2016

This is an excellent and timely review of the current state of process-based methane modeling. While other recent literature on particular methane models typically provide some brief review in the introduction and/or discussion sections, this review paper provides a very useful level of detail for understanding where, how, and why, process-based models of methane differ. As the authors note, these current methane models often poorly reproduce observed patterns, so this is an important reflective manuscript to assess the field before moving forward. That said, I do believe that the manuscript could be improved and clarified before publication. There are several relatively minor terms and phrases that require clarification that are detailed below. On a larger point, I think that it would be helpful to provide more information about representations of CH4 processes that are included within ESMs, since this is a major suggestion by the authors. They could include basic information on which models are in ESMs in Table

1, but it would also be helpful to detail plans for future representations.

Within the conclusions of their review, the authors argue that researchers should focus on the development of a fully mechanistic CH4 model that accounts for all features, and can integrate data on microbial community structure and function. There is always some tradeoff with model complexity and functionality, and I would be more convinced by the authors' conclusions that a more complex mechanistic model should be developed with all components if there was some evidence that this improves simulations over simpler representations. And furthermore, how can the increasing number of plot-to ecosystem-scale measurements of net CH4 flux be used to constrain such a complex model, except for validation? This type of very complex model would even more so require the aggregation of experimental data on microbial ecophysiology that can be used to parameterize and develop robust uncertainties for these processes, and the authors appropriately note that much of this experimental work is yet to be done. It would be helpful if the authors provided some context for understanding how much data exist to constrain these individual CH4 processes (a handful of experiments, or potentially hundreds?) and within which ecosystems. Within the section on model-data integration, I also think that it would be useful for the authors to provide more specific detail regarding ways to integrate these different data types (from net CH4 flux data to process-based experimental data).

Line by line comments follow:

L102-109: I'm confused about the reference number for the percentages: is it the percent of total carbon respiration? Or percent of total methane produced?

L235: You should also consider citing Matthews & Fung (1987) in this history: Matthews, E., and I. Fung (1987), Methane emission from natural wetlands: Global distribution, area, and environmental characteristics of sources, Global Biogeochem. Cycles, 1(1), 61–86, doi:10.1029/GB001i001p00061.

Table 1: Since the table is already large, I think that it would be useful to add which

models are within ESMs (and if so, which ESM) and which models were developed for particular regions/species (rice, Arctic, etc.).

L280-295: I think it would be helpful to add a bit more context for how and why these CH4 models are added into ESMs. The authors recommend that the third group be the focus to understand potential for reduction into ESM models, but what does it take to reduce a CH4 model into an ESM?

L315-330: This section is a bit hard to follow with respect to what exactly the differences are here among the models. I think that it would be useful to restructure this with a bit more of an introduction (like the environmental controls section) about the differences among the four distinct classes of substrate representation, with explicit list of the four classes before listing which model is in each class.

L345: I'm not sure what the authors mean by "dramatic bias" caused by a lack of representation, and this should be clarified.

L363: It's hard to follow the many different categories that the authors are creating, and I'm not completely sure which category three refers to as described here.

L370: It would be helpful to provide a bit more context for why Michaelis-Menten representation fails for multi-substrate, multi-consumer networks. Is it purely an equifinality problem?

L398: Unclear what "reported these individual processes" is referring to.

L479: I'm not sure what the "high range" refers to within this context.

L567: Unclear what is meant by "integrative tool" . . . for integrative assessment?

---

## Referee Comment (RC5) · Anonymous Referee #5 · 21 Apr 2016

The manuscript by Xu et al. reviews the past four decades of modeling methane emissions from terrestrial ecosystems. The authors provide a timeline and structure for assessing both the level of detail in terms of the processes represented and also in terms of how the processes are represented. Overall, the authors do a very nice job of comprehensively summarizing the current state of art in methane modeling and tracing the history of model development over the past four decades.

My main comments are : 1. The authors categorize the representation of processes into empirical to mechanistic approaches. This is rather subjective and it would be very helpful for the reader to have a section (1-2 paragraphs) describing how the authors define these terms. For example, even some of the mechanistic representation of processes rely on empirical response functions, and are thus only semi-mechanistic. In an ideal setting, what would be the definition of a purely mechanistic modeling approach?

2. Some of the descriptions of the processes are fairly vague. For example, even the description of methanogenesis is abbreviated to just mentioning "acetoclastic and hydrogenotrophic methanogenesis". Given that the authors are trying to emphasize a more mechanistic modeling approach, increasing the level of detail for each process would be helpful.

3. The discussion on substrate is particularly useful because most methane models do not consider this explicitly. Given the rise of atmospheric CO2, addressing how substrate has changed due to CO2 interactions, and what this means for modeling approaches and methane emissions is necessary to be mentioned.

4. Lastly, in the discussion for data needs, the list and ideas for integration within models is also very helpful. However, some discussion of the benchmark targets that the modeling community should aim for, and how to handle the uncertainties in benchmarks, would be very useful.

---

## Author Comment (AC1) · 19 May 2016

Reviewer #1

[We are deeply appreciated for the comments, which significantly improve the manuscript in terms of clarity and organization. Specifically, we 1) reorganized the introduction section; 2) defined the terrestrial ecosystems; 3) defined the primary CH4 processes; 4) revised the section for model purposes; and 5) addressed many other minor comments. All detailed point-by-point responses are listed below.]

General comments: This manuscript provides an overview and a synthesis of the evolution of models focusing on methane emissions from terrestrial ecosystems. The manuscript is based on a comparison among 39 methane models described in peer-reviewed articles, followed by a general synthesis that includes outlines of future chal-

lenges and directions in the field. I read the review with interest; it is a review which as far as I know has not been done before. Understanding the current state and potential future challenges of methane modelling will be of interest both to field researchers and for new modelling projects. The manuscript also has shortcomings that I think should be addressed before publication is considered. First, I find that the overall presentation can be improved for increased clarity, particularly with regards to sentence and paragraph structure. I have several examples in the specific comments below, but an overall assessment is recommended.

[We have made a substantial revision to address the shortcomings as stated. See below for the specific responses to reviewers.]

I also think the introduction could do a better job in outlining the scope of the manuscript, particularly I would favor some more specific information, e.g "first we will give an overview of the range of processes that have been considered in methane models, based on this we will classify existing models as determined by the range of processes considered. The following sections will review and synthesize how models deal specifically with methane production, consumption and transport within soils. . . . etc." I also recommend the authors to better define several key concepts in the manuscript. This would include your definition of a terrestrial ecosystem (see further comment below), and a definition of what constitutes a "primary process" with regards to methane dynamics (is this just your ranking of which processes that are more likely to have a stronger influence on the resulting emissions magnitude?).

[We have revised the manuscript accordingly. Specifically, we re-organized the last paragraph of the Introduction. The logic of the manuscript has been outlined at the end of the section. It highlights what we did for this review, which also addressed other minor comments in later section. The primary methane processes have been clearly described and listed.]

The term "terrestrial ecosystems" is particularly important for this manuscript, since it

defines the scope of the models that have been reviewed. How do you define terrestrial ecosystems? I.e. what is the distinction from aquatic ecosystems and why are not models of aquatic ecosystem methane emissions considered in this review? The review emphasizes the need to be able to estimate methane emissions at large regional to global scales, but aquatic ecosystems might (depending on your definition) have greater emissions than terrestrial ecosystems at the regional/global scale, so the omission of aquatic ecosystems is important. How do you define wetlands in terms of being terrestrial or aquatic ecosystems? The US and Canadian definitions of wetlands include open water wetlands with up to 2 m of standing water – are all these considered terrestrial in this review? Would it be considered a future challenge to extent the current models to include aquatic ecosystems, particularly streams, rivers, ponds and lakes?

[We have added text to clearly define the terrestrial ecosystems covered in this review, indicating the differences from aquatic ecosystems. The definition of wetlands is used. We agreed that future expansion of review to cover aquatic ecosystems might be an interesting research effort, while it is not the focus for our current review.]

Another topic that I do not think get sufficient attention in the manuscript relate to the diversity of goals for different models, and how that influences the choices made in the model development. In the introduction you bring up the fact that models can be developed for extrapolation to regional or global scales, or for process-level models that are developed to understand methane dynamics at the site level. The latter type of model requires information on many site-specific parameters (soil microbial community, iron and sulfur data etc etc), data which is not available for large regions. One recommendation in this manuscript it that more processes should be considered for methane model – however, for models aimed at regional to global scales this is likely to lead to highly unconstrained models since the data to run the models does not exist and is highly unlikely to be mapped. In short, I think there is a need to discuss how modelling goals will influence model development, particularly how this relates to available model

data inputs.

[We totally agree with the reviewer on the comment for the modeling purpose. Therefore, we added short paragraph to discuss model development and its association with the shift of models from mechanistic understanding to applicable model development. For the data requirements for parameterizing, and driving mechanistic models, we agree with the reviewer's comments, yet we believe more and more data will be generated and more insightful understandings are needed, which requires mechanistic models to fully understand the internal interaction and feedback between different processes.]

The issue of spatial data availability, used as model input, is also not discussed in the manuscript. It is my belief is that improved spatial data on wetland extents and wetland characteristics are likely to improve our accuracy of regional to global estimates of methane emissions (both magnitude and spatial patterns) much more than the incorporation of additional processes in the models. The use of different spatial products (wetland maps, inundation maps etc) for estimating global methane emissions is known to produce wildly different spatial patterns of regional methane emissions. I believe a discussion on how available data, and the use of available data, affect model development and modelling results deserve some attention in this review.

[We agree that accurate data might result in more reliable model output, which would be important than model development. But model development remains a critical improvement we need to work on in order to reduce uncertainties in quantifying CH4 budget. In addition, model development will likely provide guidance for experimental design, which is the core of data-model integration.]

Specific comments: P2 L37. I strongly discourage use the concept of global warming potential when dis- cussing methane emissions from wetlands. GWP are only applicable when considering "new" sources, i.e. changes in emissions, but cannot be used when evaluating sustained emissions. Wetlands have been emitting methane for millennia, thus their methane emissions have a much lesser additional impact on climate forcing at this point than would be concluded based on GWP (unless they have increased as a result of climate change or by other means). See Frolking and Roulet et al 2007 Glob Change Biol. P3 L70-72. This is a weak sentence to finish an introduction. P3 L74-76. Is it possible to reference the original sources?

[We have removed the term GWP and its usage for describing CH4 flux in the revision.]

P4 L83. What is meant by "primary CH4 processes"? How do you distinguish primary processes from other processes? Do these primary processes include the 3 methanogenesis processes, 2 methanotrophy processes, and the 7 transport mechanisms? Several of these processes, which I assume are what you consider primary processes since they are listed in the sentence after your statement on primary processes, are not discussed in regards to how they are represented in models. E.g. methylotrophic methanogenesis is only mentioned once, and is not discussed with regards to how it is considered by models. Also, of the seven transport mechanisms you only discuss ebullition, diffusion and plant-mediated transport – what are the other four processes? Overall, I think you need a better framework for how you classify the different processes, including a motivation on why some of these processes are to be considered in the review and why other are not.

[We have rewritten this section; 1) we defined the primary CH4 processes; 2) we re-organized the detailed CH4 processes section. We organized them into two methanogenesis, two methanotropy and three transport processes. Each transport process is composed of one or more mechanisms. The CH4 transport is discussed at higher level of three transport mechanisms. See primary CH4 processes section.]

P4 L85. Clumsy sentence structure, omit "(depending on how one counts)".

[We have removed the phrase as suggested.]

P4 L87. Importance in time and space – and you should probably highlight that it varies

by wetland characteristics.

[Revised as suggested.]

P4 L92. Perhaps a brief description is needed that explains the differences between acetoclastic and hydrogenotrophic processes, in terms of under what conditions they are more likely to dominate and why.

[We added explanations of acetoclastic and hydrogenotrophic processes.]

P4 L107. This is a awkward way of saying that upland soils are net sinks of atmospheric methane.

[We revised this sentence to emphasize that it is a range not exactly 100% for all upland. Yet we still kept ∼100% to make it consistent with other sentences. The percentage is used to help understand how much each single mechanism contribute to the total production, oxidation, or transport processes.]

P4-5 L09-15. This sentence is very long and introduced several new concepts not previously described.

[The sentence has been reorganized into two sentences. Meanwhile, we define the newly added terms diffusive and advective transports.]

P5 L16. This is the third time I have seen the same point being raised already - "process vary significantly depending on temporal and spatial scales".

[This sentence has been removed to reduce redundancies.]

P5 L17. How do you define direct and indirect effects with regards to wetland methane emissions? It is not clear to me given the examples brought up. Is the classification of direct and indirect processes different from that of primary and other processes introduced earlier?

[In our manuscript, we have information that the direct and indirect impacts are based

on their associations with CH4 production, oxidation, or transport processes.]

P5 L35. "Water sediments", do you mean "Aquatic sediments"?

[We changed it to freshwater sediment, to keep consistent with original publication.]

P5-P6 L36-57. I'm not sure this listing of the different methane models is effective. I would recommend merging this section with the section below (L181-199) on the different groups of model, i.e. to bring these models up as examples of each group.

[We have significantly shortened the paragraph by removing more than half of the listed models. And this section has been merged with the following section as suggested.]

P7 L66. What is your definition of regional simulation capability? This has not been presented.

[We added text to define the regional simulation capability. The models are defined with regional simulation capability if models directly read in and produce spatial maps.]

P8 L09. Do you have any field data that can support your statement that substrate characterization is key for modelling methane production?

[We added one citation to support our statement of strong control of substrate on methanogenesis.]

P10 L62. Unclear if you mean the third group of the three groups described in the "Model Classification' section or the third group described in the "Methanogenesis" section. I would recommend separating the models into groups once, rather than a new division of models in each section.

[We have revised the manuscript to clearly describe how we separated models based on model representation of CH4 processes. The section describing groups for methanogenesis has been reorganized as model algorithms.]

P10 L264. Seems appropriate to discuss substrate limitation and Michelis-Menten

dynamics of methanogenesis in the section on methanogenesis rather than methan-
otrophy.

[We have separated that section and moved the discussion of Michelis-Menten func-
tion into methanogenesis section, while keeping the discussion on methanotrophy in
original section as appropriate.]

P17, L43. This sentence has poor structure, also, what is meant by "was not included
in any of the three groups because that effort will likely be achieve over the long term"?

[We have revised this sentence for the purpose of improved clarity.]

P19, L09. Can you give examples of less-studies ecosystems?

[We added one sentence to show that the Arctic tundra ecosystem is an important
contributor to global CH4 budget but long-term datasets of CH4 flux are lacking.]

 P20, L31. Sentence structure: "integration between model development and data
collection is much stronger for advancing science", do you mean that integration is
important for advancing our scientific understanding of methane dynamics?

[We agree with reviewer. Yet we would like to keep sentence as it was because that
sentence is used for general scientific studies. Meanwhile, we added a detailed de-
scription of data-model integration for CH4 cycling in the following sentences.]

---

## Author Comment (AC2) · 19 May 2016

[We really appreciate the reviewer for the comments, which significantly improve the manuscript in terms of clarity and organization. Specifically, we 1) removed redundancies; 2) emphasized the importance of spatial maps of wetland data; and 3) addressed many other minor comments. All detailed point-by-point responses are listed below.]

General comments In this manuscript, the authors reviewed 39 terrestrial methane models and discussed their limitations and future opportunities. This kind of model review has been partly conducted in introduction of model intercomparison project (e.g., WETCHIMP; Melton et al., 2013, Wania et al., 2013), but I agree that this manuscript gives a more thorough overview. The 39 models were classified into several categories (or generations) from the points of processes and complexity. Also, the authors gave

good overview of underlying mechanisms of methane production, consumption, and transportation. In the light of its importance as the second important anthropogenic greenhouse gas, this manuscript is timely and within the scope of the journal.

[We appreciate the positive comments.]

The manuscript is fairly prepared, but I have several recommendations. First, I felt redundancies in the manuscript. For example, influential factors of methane processes are similarly listed in Page 5 Line 118 and Page 12 Line 322. I recommend refining the manuscript by reducing redundancies. Second, I recommend giving a broader picture of terrestrial models that include methane processes. The authors mentioned that methane schemes would be implemented into Earth system models (ESMs). Similarly, integrated terrestrial models (other than ESMs) should include methane processes to evaluate e.g. the effect of mitigation practices. Overall, I recommend that the manuscript be worth publication after moderate to major revision.

[We have carefully revised the manuscript and removed redundancies. We have also added a paragraph to discuss the implementations of CH4 module in ESMs.]

Specific comments Page 3 Line 65 This manuscript does not cover several quantitatively important processes such as methane emissions from biomass burning, termites, and ruminants. Please justify here for ignorance of these processes.

[We totally agree that CH4 emissions from biomass burning, termites, and ruminants are important. While important, these processes have not been included in this manuscript because they are not the focus of this paper.]

Page 5 Line 133 In the 1980s, E. Mattews and I. Fung (1987) achieved a pioneering work in which not only terrestrial but also atmospheric methane dynamics were simulated at the global scale. I think that their work should be mentioned in text.

[We acknowledge this pioneering work, although we did not include it because the approach in their paper is simply multiplying wetland area with measured CH4 fluxes.

[Figure]

**BGD**

It is not a modeling approach as we described. In this revision, we did cite this important work but did not treat as an independent ecosystem CH4 model.]

Page 6 Line 159 In Figure 6 of Wania et al. (2013), estimations of methane production area in the contemporary models are well summarized.

[In the revised manuscript, we added text to emphasize the importance of spatial maps of wetland distribution, and acknowledge the review of CH4 production area has been done for a group of models in Wania et al. (2013).]

Page 7 Line 190  Can you give several examples for the second group model?

[We have added few model examples as suggested.]

Page 8 Line 193  Can you give several examples for the third group model?

[We have added model examples as suggested.]

Page 9 Line 233 Can you show the 31 models by adding a column in Table 1? Page 9 Line 244 "address" should be "addressed".

[We appreciate the comment, yet we did not add it as a new column because the information has been shown in the Table 2 in a different format.]

Page 10 Line 246 and Table 1. In addition to Ridgwell et al. (1999), several methane oxidation models have been presented and could be mentioned here: e.g., Del Grosso et al. (2000) and Curry (2007).

[We do have DAYCNET, CLASS models reviewed and summarized in the Table 1.]

Page 10 Line 251 Can you indicate a typical value of the contribution of anaerobic methane oxidation in total oxidation?

[We did have this rough estimates in the primary CH4 processes section.]

Page 11 Line 275 In terms of the modeling of vertical profile, parameterization of methane diffusion coefficient within soil is critically important. Do you agree?

[We totally agree that diffusion parameter is very important in terms of simulating vertical profile of the biogeochemical processes and CH4 flux. Yet it is not focus of current review as current paper emphasizes model structure and mechanisms. We did discuss this important parameter in our revision.]

Page 13 Line 35 Yvon-Durocher et al. (2014) implied that the temperature response of methane emission would be evaluated using a single consistent model. If correct, the divergence in present models would be largely reduced. Do you agree?

[We would agree that Yvon-Durocher et al's approach is applicable for single CH4 process. Since the observed CH4 flux is a combination of many different processes. Using a single consistent model might not be the best way to represent CH4 flux. Yvon-Durocher's approach provides a theoretical understanding of some consistencies between observed CH4 fluxes across space.]

Page 14 Line 356 As long as I know, only a few global dataset of soil pH is available. Also, in situ measurement and model prediction of soil pH are rather difficult. I think these difficulties in using soil pH should be noted.

[We agree that global dataset of soil pH is lacking, yet a number of field experiments and modeling studies do confirm the importance of soil pH to CH4 flux. We did note the difficulties for modeling soil pH in the revision.]

Page 15 Line 380 It looks wired to give a summary at this place, because it is usually given at the end of the manuscript. Actually, the statements around Page 16 Line 411 are as if your conclusion.

[This summary section is a short paragraph for CH4 modeling section only, while the last conclusion section is for high-level summary and key findings for the whole manuscript. We would still keep this section but make it as a sub-section of modeling section.]

Page 18 Line 460 A few more processes not mentioned here have been presented:

e.g., emission from tank bromeliads (Martinson et al., 2010) and emission from small ponds (Holgerson and Raymond, 2016).

[We have included these new findings in the manuscript and identified them as a knowledge gap and future direction for modeling community.]

Page 19 Line 504 I recommend adding one more (6th?) challenge. Modeling of human-natural processes such as emission from managed ponds and estuaries is important in terms of mitigation. Namely, we should consider both natural biogeochemical processes and human management effects.

[We have added it in the revised manuscript as suggested. We appreciated the reviewer for pointing this out.]

Page 21 Line 540
Do you mean "Markov Chain Monte Carlo (MCMC)"?

[Mistake corrected.]

Page 25 Line 623
Please correct information for Bohn et al. (2015):

[Mistake corrected, thanks.]

Figure 4 Can you include the microbial community factor into the figure?

[We have revised the figure to show several different functional groups of microbes that control the CH4 processes.]

---

## Author Comment (AC3) · 19 May 2016

[We really appreciate the comments, which significantly improve the manuscript in terms of clarity and organization. All detailed point-by-point responses are listed below.]

Overall Evaluation This manuscript presents a review of approaches used to model methane dynamics in terrestrial ecosystems in the last four decades. The review largely focuses on describing the variability in structure and mathematical descriptions of processes among 39 terrestrial methane models. Parameterization issues are touched upon in the section on environmental controls, mostly with respect to variability in Q10 (which affects temperature sensitivity of processes). The discussion makes suggestions for adding more complexity to methane models, primarily along the lines

of more explicitly considering microbial processes and dynamics. The discussion finishes with identifying knowledge gaps, modeling challenges, data needs, and the need for data-model integration.

[We appreciated the positive comments.]

This manuscript tries to cover a lot of ground. The primary strength of the manuscript, in my opinion, is largely in the description of variability in mathematical descriptions of processes. The other aspects of the review didn't provide a lot insight in my opinion, as the issues discussed were in many cases just touched upon and were not well developed. My main concern about this manuscript is that in trying to cover a lot of ground, it covers some of that ground poorly. I think there are several issues to address to improve the review. First, I think there are some general organization issues that could be addressed to improve the manuscript. Second, there are a number of cases in the presentation of putting the "cart" before the "horse". Third, I didn't find that the description in the variability in structure (as depicted in Figure 3) was based on an objective evaluation of the 39 terrestrial models. Fourth, there a number of assertions in the manuscript that should be presented as more open issues. Fifth, the challenge of scaling is only touched upon in the manuscript and needs to be better developed, and there is a need for some discussion of reconciliation with atmospheric data analyses. Sixth, beside the scaling/reconciliation issue, I also found several issues that need to be better developed/discussed including the modeling of ebullition, vertical representation of processes, model benchmarking, and data-model integration. Below I go into more depth on each of these issues, and finish my review with a listing of specific comments.

[We have made a substantial revision to address the comments. Specifically, (1) we reorganized several sections to make them clearer, particularly the modeling section; (2) we clearly revised some statements to make them more consistent with the results; (3) scaling is not the key focus of this review, therefore, we did not expand writing on scaling and its reconciliation with atmospheric data analysis; yet we did emphasize that satellite data of atmospheric CH4 concentration could be used for model validation; (4)

we added a section for discussing presenting CH4 module in ESMs.]

Issue 1: Organizational issues in the manuscript. The manuscript starts out well, but then gradually gets more and more disorganized. There is a lot of overlap of material between some of the later sections of the manuscript that could be eliminated with a more effective organization. Perhaps consider the organization of Luo et al. (2016, Global Biogeochemical Cycles), which review soil carbon models. The organization of that paper is (1) model structure, (2) model parameterization, and (3) external forcing. I think additional in this manuscript concerns scaling and reconciliation with atmospheric data. The strength of this manuscript is that it generally does a good job of reviewing model structure, but a rather inadequate job of reviewing model parameterization, external forcing, scaling, and reconciliation issues.

[We did remove some redundancies in different sections. This manuscript is not designed to cover model parameterization and external forcing specifically. Those two sections were discussed briefly model development perspective.]

Issue 2: "Cart" before the "Horse" issues. There are a number of places in the manuscript where the "cart" comes before the "horse", from the perspective of this being a review paper. For example, the citation to Figure 2 on line 162 talks about the timeline for inclusion of "key mechanisms", but these mechanisms haven't been described in a general sense yet. Table 2, which contains the list of "key mechanisms" isn't cited until line 175. Even when Table 2 is cited, the general reader gets no background on these mechanisms/features of models, as it is not used beyond a simple citation at the end of a sentence. Other rough spots in the manuscript involve adequately describing terms used in the manuscript. For example, acetoclastic and hydrogenotrophic methanogenesis suddenly appears on lines 238-240 without any prior description. "Advective transport" (line 203) is also not described.

[For the citation to Figure 2, although those processes and their representation in models have not been reviewed, yet the processes themselves have been reviewed in

"primary CH4 processes" section. For the acetoclastic and hydrogenotrophic methano-genesis, we have included detailed definitions at their early occurrences. The advective transport has been defined as well.]

Issue 3: Analysis of the variability in structure. What is the basis for defining three different types of models? It seems to me that this could be done in a much more objective fashion by doing some sort of cluster analysis among the 39 models reviewed in this study. Information from Tables 1, 2, and 3 could be put into an objective cluster analysis so that we better understood what factors seem to cause models to be distinct (or not distinct) from each other.

[We really appreciated the suggestion of doing a cluster analysis. We did a cluster analysis based on model characteristics of representation of methanogenesis processes, methanotrophy processes, transport pathways, oxygen availability, multiple soil layers etc. All 40 models could be classified into three groups, which is consistent with our previous classification. See the updated Figure 3 and relevant text.]

Issue 4: There are a number of assertions in the manuscript that have not been justified by any sort of rational analysis/argument. For example, why make a recommendation in the last sentence of section 4 (lines 196-198) on the third types of models as the means of moving forward with respect to improving reduced form models for application in Earth System Model applications? First of all, this is too early in the manuscript. Second, doesn't making this recommendation conflict with the sentence on lines 211-212 that the optimum complexity remains to be determined? At the end of section 6 there are four recommendations for models "based on the above-mentioned needs" and a citation to Figure 4. I didn't find the previous text in section 6 as being very helpful for establishing these as the top needs. This all comes before the section 7, which talks about knowledge gaps and data needs. The arrows for benchmarking and data assimilation in Figure 4 have not been developed, and the issues of vertical trans-port/diffusion have only been touched upon. Also, the top recommendation that "the models (features?) should be embedded in an Earth System Model" seems strange to

make here. The point here is that arguments have not been well enough organized and crafted to effectively make these recommendations. This sort of all gets back to issues 1 and 2 above. Finally, I can't say that I'm very fond of Figure 4 as being the synthetic figure for this manuscript – we've seen a lot of these sort of figures over the years. I suggest thinking about something that is truly synthetic based on this manuscript.

[We have revised the manuscript to address all comments. The recommendation of the third group of model in the early section of the paper has been removed as suggested. Then it is not in conflict with later section as reviewer suggested. We added description and summary of CH4 model representation in ESMs, the model classification have been done in a mathematical way – a cluster analysis. Other writing issues have been addressed as well. The original Figure 4 is a framework showing future model development as we envisioned, it is combination of summarized and visionary framework. Although it is little lack of evidence, we do believe it will be the key direction for CH4 model development and application.]

Issue 5: The issues of scaling and reconciliation with atmospheric data. Scaling is an important issue. It does pop up several places in the manuscript as a sort of "between the lines" issue, but it really needs its own section. I also think that the issue of reconciling model applications at particular scales with data from atmospheric analyses needs to be part of the discussion.

[Since scaling and reconciliation are not the key focus of this manuscript, we did not plan to expand that section in this revision.]

Issue 6: Other issues. I also found several issues that need to be better developed/discussed including the modeling of ebullition, vertical representation of processes, model benchmarking, and data-model integration. For example, transport mechanisms don't even show up as key features in Table 2, although they do appear somewhat in Table 1. These issues are touched upon in several places in the manuscript, but are not really effectively dealt with in a meaningful way.

[We have added more text and wordings to make those statements strong and solid.]

Specific comments Line 104-105: "contributes" is not really the right verb to use here. Just says "varies from 1 to 90%", for example.

[We still keep "contributes" because it emphasizes the contribution of individual process to the total oxidation or production.]

Line 106-107: I really don't know what you mean by "oxidation of atmospheric CH4 contributes". Aren't all of the previous mechanisms in this paragraph ultimately oxidation of atmospheric CH4, albeit in the open pore space of the soil.

[It emphasizes the oxidation of atmospheric CH4, taking up CH4 from atmosphere. This process is defined to distinguish from oxidation of CH4 produced from soils.]

Line 109: Perhaps start a new paragraph after "methanotrophy.".

[We separated it as a new paragraph.]

Lines 109-116: There is no information for the uninitiated reader to understand how these pathways differ from each other.

[We added one small paragraph to define different transport pathways.]

Line 120: I think this might be the only occurrence of "wind speed" in the manuscript. What do you mean by "wind speed" as an environmental factor.

[We revised the manuscript to have a bit more description of wind speed impacts on CH4 flux.]

Line 121: Define what you mean by "indirect" vs. "direct" environmental factors.

[We revised the manuscript to define the direct and indirect environmental factors.]

Line 147: I don't think Fan et al. (2013, Peatland DOS-TEM) has anything to do with the Zhuang et al. (2014) model in that it has a number of different features and to my understanding the two models do not share any code base.

[We have confirmed with Dr. Zhaosheng Fan, and treated the DOS-TEM as another independent CH4 model in the revision.]

Line 162: As mentioned earlier, the reader needs to know more about the key mechanisms before you present/interpret Figure 2.

[We have added definition for some key mechanisms in the manuscript.]

Line 175: Need to make better use of Table 2 in the manuscript. As I indicated earlier, transport mechanisms need to be included in Table 2.

[We expanded the Table 2 to include the model information on CH4 transport pathway.]

Line 213: Does use of "first group of models" refer to model types in Figure 3, or to the first set of empirical models referred to in the first paragraph of section 4.1?

[We have revised the manuscript to be clearer on this issue. The CH4 models were classified as groups, while methangoenesis was categorized as model algorithms.]

Line 238-240: Where does the information on acetoclastic and hydrogenotropic methanogenesis appear in Table 3? Note that these production processes have not been defined for the reader.

[We have added definitions for acetoclastic and hydrogenotropic methanogenesis in the revised manuscript.]

Line 280: Why is Zhuang (2004) cited here in the context of immediately transporting CH4? This model is primarily a monthly model with a pseudo-daily time step. This transport issue is an important temporal scaling issue, and one which should appear in a separate section on temporal scaling.

[Thanks for pointing out this inappropriate expression. We have removed citation of Zhuang (2004), and added another model as an example.]

Line 286: I think you should change "will likely" to "can". Line 287: I think you should

change "impossible" to "not straight forward".

[Revised as suggested.]

Line 291: I note that ebullition is not adequately treated in this section (section 4.4).

[We do have ebullition in the section. In the revision, we have revised the section to have more specific information for ebullition.]

Line 292: Why is this the "final" bottleneck, or why is even referred to as a "bottleneck". Line 303: Define advective transport.

[We revised it to bottleneck, and added definition of advective transport.]

Line 313: I think you should change "most" to "some". Note that ebullition seems to be ignored in these three "transport" challenges. It is a dominant pathway in some systems.

[Revised as suggested.]

Line 319: I note that the simulation of variability in some environmental controls is not adequately treated in section 4.5 on environmental controls.

[We have revised the section to better describe variability in environmental controls.]

Lines 331-332: I think that this sentence needs to refer to Eq 9, 10, and 11 instead of 10, 11, and 12. Note that the third function in Eq 9 is essentially equivalent to Eq 10 in that the Q10 can be derived from the exponent.

[Mistake corrected.]

Line 347: I think you mean Eqs. 13-16 instead of 12-15.

[Mistake corrected.]

Lines 356-367: Do any models represent pH variability in time? It would be useful to know how models represent pH variability in space.

[Figure]

[We agree that pH variability is important and only few models consider dynamics of pH in soil over time and across space. Due to recent studies suggesting the importance of pH on CH4 flux, it would be noteworthy to point out its importance for future model development.]

Lines 393-394: Why is the comparison of high frequency observational data needed for future model-model inter-comparison? I think it would be most important to high quality seasonal and interannual estimates derived from observations to effectively test and compare models.

[We have revised to reflect this point.]

Line 405: With respect to shifts, are you referring to shifts in time or in space?

[We have revised to clarify it is temporal shifts.]

Line 479: What do you mean by "order 1-10". Do you mean by a "factor of 1-10"? The language could be confused for "orders of magnitude".

[We have revised it to a factor of 1-10.]

---

## Author Comment (AC4) · 19 May 2016

[We really appreciate the reviewer for the comments, which significantly improve the manuscript in terms of clarity and organization. All detailed point-by-point responses are listed below.]

This is an excellent and timely review of the current state of process-based methane modeling. While other recent literature on particular methane models typically provide some brief review in the introduction and/or discussion sections, this review paper provides a very useful level of detail for understanding where, how, and why, process-based models of methane differ. As the authors note, these current methane models often poorly reproduce observed patterns, so this is an important reflective manuscript to assess the field before moving forward. That said, I do believe that the manuscript

could be improved and clarified before publication. There are several relatively minor terms and phrases that require clarification that are detailed below. On a larger point, I think that it would be helpful to provide more information about representations of CH4 processes that are included within ESMs, since this is a major suggestion by the authors. They could include basic information on which models are in ESMs in Table 1, but it would also be helpful to detail plans for future representations.

[We agree with the description of this work, and the needs for more discussion of CH4 model representation in ESMs. We have added one paragraph to summarize how ESMs include CH4 module; what is the likely future direction for ESMs development in terms of CH4 representation.]

Within the conclusions of their review, the authors argue that researchers should focus on the development of a fully mechanistic CH4 model that accounts for all features, and can integrate data on microbial community structure and function. There is always some tradeoff with model complexity and functionality, and I would be more convinced by the authors' conclusions that a more complex mechanistic model should be developed with all components if there was some evidence that this improves simulations over simpler representations. And furthermore, how can the increasing number of plot-to ecosystem-scale measurements of net CH4 flux be used to constrain such a complex model, except for validation? This type of very complex model would even more so require the aggregation of experimental data on microbial ecophysiology that can be used to parameterize and develop robust uncertainties for these processes, and the authors appropriately note that much of this experimental work is yet to be done. It would be helpful if the authors provided some context for understanding how much data exist to constrain these individual CH4 processes (a handful of experiments, or potentially hundreds?) and within which ecosystems. Within the section on model-data integration, I also think that it would be useful for the authors to provide more specific detail regarding ways to integrate these different data types (from net CH4 flux data to process-based experimental data).

[We have added discussion about the tradeoff of developing a more mechanistic model and a simple empirical model. Meanwhile, the classification of empirical model and process-based model has been expanded. Meanwhile, we totally agreed that constraining mechanistically model is really challenging, yet it is becoming more and more applicable as the scientific community is expanding measurements of CH4 flux and processes, as well as developing new model optimization algorithms. For example, SPRUCE, NGEE-Arctic and NGEE-Tropic projects within DOE are taking this intensive measurements and integration with models. A new model optimization algorithm has been developed associated with CLM framework and ALM framework, we believe the mechanistic models will be more powerful in near future along with these lines of advancements.]

Line by line comments follow: L102-109: I'm confused about the reference number for the percentages: is it the percent of total carbon respiration? Or percent of total methane produced?

[Those percentage numbers emphasize specific processes to the single function of CH4 cycling. For example, acetoclastic methanogenesis contributes to ∼60-100% to the total CH4 production.]

L235: You should also consider citing Matthews & Fung (1987) in this history: Matthews, E., and I. Fung (1987), Methane emission from natural wetlands: Global distribution, area, and environmental characteristics of sources, Global Biogeochem. Cycles, 1(1), 61–86, doi:10.1029/GB001i001p00061.

[We totally agree that the pioneering work by Matthews and Fung is important and should be cited in the manuscript. We have cited it in the revised manuscript.]

Table 1: Since the table is already large, I think that it would be useful to add which models are within ESMs (and if so, which ESM) and which models were developed for particular regions/species (rice, Arctic, etc.).

[We appreciated the reviewer for pointing out this issue. The information of which CH4 models are embedded in ESMs has been summarized in the Table.]

L280-295: I think it would be helpful to add a bit more context for how and why these CH4 models are added into ESMs. The authors recommend that the third group be the focus to understand potential for reduction into ESM models, but what does it take to reduce a CH4 model into an ESM?

[We have added texts to emphasize the importance of representing CH4 module in ESMs.]

L315-330: This section is a bit hard to follow with respect to what exactly the differences are here among the models. I think that it would be useful to restructure this with a bit more of an introduction (like the environmental controls section) about the differences among the four distinct classes of substrate representation, with explicit list of the four classes before listing which model is in each class.

[This section has been re-organized little bit for clarity purpose.]

L345: I'm not sure what the authors mean by "dramatic bias" caused by a lack of representation, and this should be clarified.

[We have revised the sentence to clearly reflect the importance of representing these two mechanisms; the bias in surface CH4 flux will likely be biased if we do not represent these two mechanisms. Studies have confirmed that the surface layer and bottom layer have different mechanisms dominated CH4 production (McCalley et al., 2014), therefore, if we do not consider two mechanisms, we will not be able to simulate this shift and likely the surface fluxes caused by this function shift in response to environmental change. McCalley, C. K., Woodcroft, B. J., Hodgkins, S. B., Wehr, R. A., Kim, E.-H., Mondav, R., Crill, P. M., Chanton, J. P., Rich, V. I., Tyson, G. W., and Saleska, S. R.: Methane dynamics regulated by microbial community response to permafrost thaw, Nature, 514, 478-481, 2014.]

[Figure]

L363: It's hard to follow the many different categories that the authors are creating, and I'm not completely sure which category three refers to as described here.

[We have revised the manuscript. The three groups of CH4 models are remained, while we changed four groups of methanogenesis to four modeling algorithms for methanogenesis. The classification of three groups of CH4 models have been demonstrated with a cluster analysis as suggested by another reviewer (new Figure 3).]

L370: It would be helpful to provide a bit more context for why Michaelis-Menten representation fails for multi-substrate, multi-consumer networks. Is it purely an equifinality problem?

[We rewrote this sentence to acknowledge the new approach developed by Riley's group. The ECA approach might be good for multi-substrate, multi-consumer biogeochemistry reaction network. We added a short description in this aspect.]

L398: Unclear what "reported these individual processes" is referring to.
L479: I'm not sure what the "high range" refers to within this context.

[L398 emphasizes the individual processes discussed in previous section. While L479 primarily focuses on processes caused hot spot and hot moments in CH4 flux. In the revised manuscript, we revised those two sentences for clarity purpose.]

L567: Unclear what is meant by "integrative tool" . . . for integrative assessment?

[We used "integrative tool" to emphasize that the model can be used to integrate multiple sources of data to reach a better understanding of the system and better budget quantification.]

---

## Author Comment (AC5) · 19 May 2016

[We really appreciate the reviewer for the comments, which significantly improve the manuscript in terms of clarity and organization. All detailed point-by-point responses are listed below.]

The manuscript by Xu et al. reviews the past four decades of modeling methane emissions from terrestrial ecosystems. The authors provide a timeline and structure for assessing both the level of detail in terms of the processes represented and also in terms of how the processes are represented. Overall, the authors do a very nice job of comprehensively summarizing the current state of art in methane modeling and tracing the history of model development over the past four decades.

[We appreciate the positive comments.]

[Figure]

My main comments are : 1. The authors categorize the representation of processes into empirical to mechanistic approaches. This is rather subjective and it would be very helpful for the reader to have a section (1-2 paragraphs) describing how the authors define these terms. For example, even some of the mechanistic representation of processes rely on empirical response functions, and are thus only semi-mechanistic. In an ideal setting, what would be the definition of a purely mechanistic modeling approach?

[We totally agree that separation of mechanistic and empirical is rather arbitrary, while it does help understand the model representation of CH4 processes. In this revised manuscript, we provideddetailed description to show how to define the empirical and mechanistic models in terms of modeling CH4 dynamics.]

2. Some of the descriptions of the processes are fairly vague. For example, even the description of methanogenesis is abbreviated to just mentioning "acetoclastic and hydrogenotrophic methanogenesis". Given that the authors are trying to emphasize a more mechanistic modeling approach, increasing the level of detail for each process would be helpful.

[We agree that it is important to have more detailed description of two methanogenesis processes. Yet the processes themselves have been reviewed, while for all the present CH4 model, only few models simulate acetoclastic and hydrogenotrophic methanogenesis; which are not sufficient for a detailed review section.]

3. The discussion on substrate is particularly useful because most methane models do not consider this explicitly. Given the rise of atmospheric CO2, addressing how substrate has changed due to CO2 interactions, and what this means for modeling approaches and methane emissions is necessary to be mentioned.

[We added a short paragraph to discuss the potential impacts of elevated CO2 and substrate on CH4 emission.]

4. Lastly, in the discussion for data needs, the list and ideas for integration within

models is also very helpful. However, some discussion of the benchmark targets that the modeling community should aim for, and how to handle the uncertainties in benchmarks, would be very useful.

[We have added one small paragraph to summarize the benchmarking targets of the benchmarking system and uncertainties in benchmark.]

---

## Author Comment (AC6) · 19 May 2016

Reviewer #1

*[We are deeply appreciated for the comments, which significantly improve the manuscript in terms of clarity and organization. Specifically, we 1) reorganized the introduction section; 2) defined the terrestrial ecosystems; 3) defined the primary $CH_4$ processes; 4) revised the section for model purposes; and 5) addressed many other minor comments. All detailed point-by-point responses are listed below.]*

General comments: This manuscript provides an overview and a synthesis of the evolution of models focusing on methane emissions from terrestrial ecosystems. The manuscript is based on a comparison among 39 methane models described in peer- reviewed articles, followed by a general synthesis that includes outlines of future challenges and directions in the field. I read the review with interest; it is a review which as far as I know has not been done before. Understanding the current state and potential future challenges of methane modelling will be of interest both to field researchers and for new modelling projects. The manuscript also has shortcomings that I think should be addressed before publication is considered. First, I find that the overall presentation can be improved for increased clarity, particularly with regards to sentence and paragraph structure. I have several examples in the specific comments below, but an overall assessment is recommended.

*[We have made a substantial revision to address the shortcomings as stated. See below for the specific responses to reviewers.]*

I also think the introduction could do a better job in outlining the scope of the manuscript, particularly I would favor some more specific information, e.g "first we will give an overview of the range of processes that have been considered in methane models, based on this we will classify existing models as determined by the range of processes considered. The following sections will review and synthesize how models deal specifically with methane production, consumption and transport within soils. . . . etc." I also recommend the authors to better define several key concepts in the manuscript. This would include your definition of a terrestrial ecosystem (see further comment below), and a definition of what constitutes a "primary

process" with regards to methane dynamics (is this just your ranking of which processes that are more likely to have a stronger influence on the resulting emissions magnitude?).

*[We have revised the manuscript accordingly. Specifically, we re-organized the last paragraph of the Introduction. The logic of the manuscript has been outlined at the end of the section. It highlights what we did for this review, which also addressed other minor comments in later section. The primary methane processes have been clearly described and listed.]*

The term "terrestrial ecosystems" is particularly important for this manuscript, since it defines the scope of the models that have been reviewed. How do you define terrestrial ecosystems? I.e. what is the distinction from aquatic ecosystems and why are not models of aquatic ecosystem methane emissions considered in this review? The review emphasizes the need to be able to estimate methane emissions at large regional to global scales, but aquatic ecosystems might (depending on your definition) have greater emissions than terrestrial ecosystems at the regional/global scale, so the omission of aquatic ecosystems is important. How do you define wetlands in terms of being terrestrial or aquatic ecosystems? The US and Canadian definitions of wetlands include open water wetlands with up to 2 m of standing water – are all these considered terrestrial in this review? Would it be considered a future challenge to extent the current models to include aquatic ecosystems, particularly streams, rivers, ponds and lakes?

*[We have added text to clearly define the terrestrial ecosystems covered in this review, indicating the differences from aquatic ecosystems. The definition of wetlands is used. We agreed that future expansion of review to cover aquatic ecosystems might be an interesting research effort, while it is not the focus for our current review.]*

Another topic that I do not think get sufficient attention in the manuscript relate to the diversity of goals for different models, and how that influences the choices made in the model development. In the introduction you bring up

the fact that models can be developed for extrapolation to regional or global scales, or for process-level models that are developed to understand methane dynamics at the site level. The latter type of model requires information on many site-specific parameters (soil microbial community, iron and sulfur data etc etc), data which is not available for large regions. One recommendation in this manuscript it that more processes should be considered for methane model – however, for models aimed at regional to global scales this is likely to lead to highly unconstrained models since the data to run the models does not exist and is highly unlikely to be mapped. In short, I think there is a need to discuss how modelling goals will influence model development, particularly how this relates to available model data inputs.

*[We totally agree with the reviewer on the comment for the modeling purpose. Therefore, we added short paragraph to discuss model development and its association with the shift of models from mechanistic understanding to applicable model development. For the data requirements for parameterizing, and driving mechanistic models, we agree with the reviewer's comments, yet we believe more and more data will be generated and more insightful understandings are needed, which requires mechanistic models to fully understand the internal interaction and feedback between different processes.]*

The issue of spatial data availability, used as model input, is also not discussed in the manuscript. It is my belief is that improved spatial data on wetland extents and wet- land characteristics are likely to improve our accuracy of regional to global estimates of methane emissions (both magnitude and spatial patterns) much more than the in- corporation of additional processes in the models. The use of different spatial products (wetland maps, inundation maps etc) for estimating global methane emissions is known to produce wildly different spatial patterns of regional methane emissions. I believe a discussion on how available data, and the use of available data, affect model development and modelling results deserve some attention in this review.

*[We agree that accurate data might result in more reliable model output, which would be important than model development. But model development*

*remains a critical improvement we need to work on in order to reduce uncertainties in quantifying $CH_4$ budget. In addition, model development will likely provide guidance for experimental design, which is the core of data-model integration.]*

Specific comments:

P2 L37. I strongly discourage use the concept of global warming potential when dis- cussing methane emissions from wetlands. GWP are only applicable when considering "new" sources, i.e. changes in emissions, but cannot be used when evaluating sustained emissions. Wetlands have been emitting methane for millennia, thus their methane emissions have a much lesser additional impact on climate forcing at this point than would be concluded based on GWP (unless they have increased as a result of climate change or by other means). See Frolking and Roulet et al 2007 Glob Change Biol.P3 L70-72. This is a weak sentence to finish an introduction. P3 L74-76. Is it possible to reference the original sources?

*[We have removed the term GWP and its usage for describing $CH_4$ flux in the revision.]*

P4 L83. What is meant by "primary CH4 processes"? How do you distinguish primary processes from other processes? Do these primary processes include the 3 methanogenesis processes, 2 methanotrophy processes, and the 7 transport mechanisms? Several of these processes, which I assume are what you consider primary processes since they are listed in the sentence after your statement on primary processes, are not discussed in regards to how they are represented in models. E.g. methylotrophic methanogenesis is only mentioned once, and is not discussed with regards to how it is considered by models. Also, of the seven transport mechanisms you only discuss ebullition, diffusion and plant-mediated transport – what are the other four processes? Overall, I think you need a better framework for how you classify the different processes, including a motivation on why some of these processes are to be considered in the review and why other are not.

*[We have rewritten this section; 1) we defined the primary $CH_4$ processes; 2) we reorganized the detailed $CH_4$ processes section. We organized them into two methanogenesis, two methanotropy and three transport processes. Each transport process is composed of one or more mechanisms. The $CH_4$ transport is discussed at higher level of three transport mechanisms. See primary $CH_4$ processes section.]*

P4 L85. Clumsy sentence structure, omit "(depending on how one counts)".

*[We have removed the phrase as suggested.]*

P4 L87. Importance in time and space – and you should probably highlight that it varies by wetland characteristics.

*[Revised as suggested.]*

P4 L92. Perhaps a brief description is needed that explains the differences between acetoclastic and hydrogenotrophic processes, in terms of under what conditions they are more likely to dominate and why.

*[We added explanations of acetoclastic and hydrogenotrophic processes.]*

P4 L107. This is a awkward way of saying that upland soils are net sinks of atmospheric methane.

*[We revised this sentence to emphasize that it is a range not exactly 100% for all upland. Yet we still kept ~100% to make it consistent with other sentences. The percentage is used to help understand how much each single mechanism contribute to the total production, oxidation, or transport processes.]*

P4-5 L09-15. This sentence is very long and introduced several new concepts

not previously described.

*[The sentence has been reorganized into two sentences. Meanwhile, we define the newly added terms diffusive and advective transports.]*

P5 L16. This is the third time I have seen the same point being raised already - "process vary significantly depending on temporal and spatial scales".

*[This sentence has been removed to reduce redundancies.]*

P5 L17. How do you define direct and indirect effects with regards to wetland methane emissions? It is not clear to me given the examples brought up. Is the classification of direct and indirect processes different from that of primary and other processes introduced earlier?

*[In our manuscript, we have information that the direct and indirect impacts are based on their associations with $CH_4$ production, oxidation, or transport processes.]*

P5 L35. "Water sediments", do you mean "Aquatic sediments"?

*[We changed it to freshwater sediment, to keep consistent with original publication.]*

P5-P6 L36-57. I'm not sure this listing of the different methane models is effective. I would recommend merging this section with the section below (L181-199) on the different groups of model, i.e. to bring these models up as examples of each group.

*[We have significantly shortened the paragraph by removing more than half of the listed models. And this section has been merged with the following section as suggested.]*

P7 L66. What is your definition of regional simulation capability? This has not been presented.

*[We added text to define the regional simulation capability. The models are defined with regional simulation capability if models directly read in and produce spatial maps.]*

P8 L09. Do you have any field data that can support your statement that substrate characterization is key for modelling methane production?

*[We added one citation to support our statement of strong control of substrate on methanogenesis.]*

P10 L62. Unclear if you mean the third group of the three groups described in the "Model Classification' section or the third group described in the "Methanogenesis" section. I would recommend separating the models into groups once, rather than a new division of models in each section.

*[We have revised the manuscript to clearly describe how we separated models based on model representation of $CH_4$ processes. The section describing groups for methanogenesis has been reorganized as model algorithms.]*

P10 L264. Seems appropriate to discuss substrate limitation and Michelis-Menten dynamics of methanogenesis in the section on methanogenesis rather than methan- otrophy.

*[We have separated that section and moved the discussion of Michelis-Menten function into methanogenesis section, while keeping the discussion on methanotrophy in original section as appropriate.]*

P17, L43. This sentence has poor structure, also, what is meant by "was not included in any of the three groups because that effort will likely be achieve over the long term"?

*[We have revised this sentence for the purpose of improved clarity.]*

P19, L09. Can you give examples of less-studies ecosystems?

*[We added one sentence to show that the Arctic tundra ecosystem is an important contributor to global $CH_4$ budget but long-term datasets of $CH_4$ flux are lacking.]*

P20, L31. Sentence structure: "integration between model development and data collection is much stronger for advancing science", do you mean that integration is important for advancing our scientific understanding of methane dynamics?

*[We agree with reviewer. Yet we would like to keep sentence as it was because that sentence is used for general scientific studies. Meanwhile, we added a detailed description of data-model integration for $CH_4$ cycling in the following sentences.]*

Reviewer #2

*[We really appreciate the reviewer for the comments, which significantly improve the manuscript in terms of clarity and organization. Specifically, we 1) removed redundancies; 2) emphasized the importance of spatial maps of wetland data; and 3) addressed many other minor comments. All detailed point-by-point responses are listed below.]*

General comments

In this manuscript, the authors reviewed 39 terrestrial methane models and discussed their limitations and future opportunities. This kind of model review has been partly conducted in introduction of model intercomparison project (e.g., WETCHIMP; Melton et al., 2013, Wania et al., 2013), but I agree that this manuscript gives a more thorough overview. The 39 models were classified into several categories (or generations) from the points of processes and complexity. Also, the authors gave good overview of underlying mechanisms of methane production, consumption, and transportation. In the light of its importance as the second important anthropogenic greenhouse gas, this manuscript is timely and within the scope of the journal.

*[We appreciate the positive comments.]*

The manuscript is fairly prepared, but I have several recommendations. First, I felt redundancies in the manuscript. For example, influential factors of methane processes are similarly listed in Page 5 Line 118 and Page 12 Line 322. I recommend refining the manuscript by reducing redundancies. Second, I recommend giving a broader picture of terrestrial models that include methane processes. The authors mentioned that methane schemes would be implemented into Earth system models (ESMs). Similarly, integrated terrestrial models (other than ESMs) should include methane processes to evaluate e.g. the effect of mitigation practices. Overall, I recommend that the manuscript be worth publication after moderate to major revision.

*[We have carefully revised the manuscript and removed redundancies. We have also added a paragraph to discuss the implementations of $CH_4$ module*

*in ESMs.]*

Specific comments

Page 3 Line 65

This manuscript does not cover several quantitatively important processes such as methane emissions from biomass burning, termites, and ruminants. Please justify here for ignorance of these processes.

*[We totally agree that $CH_4$ emissions from biomass burning, termites, and ruminants are important. While important, these processes have not been included in this manuscript because they are not the focus of this paper.]*

Page 5 Line 133

In the 1980s, E. Mattews and I. Fung (1987) achieved a pioneering work in which not only terrestrial but also atmospheric methane dynamics were simulated at the global scale. I think that their work should be mentioned in text.

*[We acknowledge this pioneering work, although we did not include it because the approach in their paper is simply multiplying wetland area with measured $CH_4$ fluxes. It is not a modeling approach as we described. In this revision, we did cite this important work but did not treat as an independent ecosystem $CH_4$ model.]*

Page 6 Line 159

In Figure 6 of Wania et al. (2013), estimations of methane production area in the contemporary models are well summarized.

*[In the revised manuscript, we added text to emphasize the importance of spatial maps of wetland distribution, and acknowledge the review of $CH_4$*

*production area has been done for a group of models in Wania et al. (2013).]*

Page 7 Line 190 Can you give several examples for the second group model?

*[We have added few model examples as suggested.]*

Page 8 Line 193 Can you give several examples for the third group model?

*[We have added model examples as suggested.]*

Page 9 Line 233Can you show the 31 models by adding a column in Table 1?
Page 9 Line 244"address" should be "addressed".

*[We appreciate the comment, yet we did not add it as a new column because the information has been shown in the Table 2 in a different format.]*

Page 10 Line 246 and Table 1. In addition to Ridgwell et al. (1999), several methane oxidation models have been presented and could be mentioned here: e.g., Del Grosso et al. (2000) and Curry (2007).

*[We do have DAYCNET, CLASS models reviewed and summarized in the Table 1.]*

Page 10 Line 251 Can you indicate a typical value of the contribution of anaerobic methane oxidation in total oxidation?

*[We did have this rough estimates in the primary $CH_4$ processes section.]*

Page 11 Line 275 In terms of the modeling of vertical profile, parameterization of methane diffusion coefficient within soil is critically

important. Do you agree?

*[We totally agree that diffusion parameter is very important in terms of simulating vertical profile of the biogeochemical processes and $CH_4$ flux. Yet it is not focus of current review as current paper emphasizes model structure and mechanisms. We did discuss this important parameter in our revision.]*

Page 13 Line 35 Yvon-Durocher et al. (2014) implied that the temperature response of methane emission would be evaluated using a single consistent model. If correct, the divergence in present models would be largely reduced. Do you agree?

*[We would agree that Yvon-Durocher et al's approach is applicable for single $CH_4$ process. Since the observed $CH_4$ flux is a combination of many different processes. Using a single consistent model might not be the best way to represent $CH_4$ flux. Yvon-Durocher's approach provides a theoretical understanding of some consistencies between observed $CH_4$ fluxes across space.]*

Page 14 Line 356 As long as I know, only a few global dataset of soil pH is available. Also, in situ measurement and model prediction of soil pH are rather difficult. I think these difficulties in using soil pH should be noted.

*[We agree that global dataset of soil pH is lacking, yet a number of field experiments and modeling studies do confirm the importance of soil pH to $CH_4$ flux. We did note the difficulties for modeling soil pH in the revision.]*

Page 15 Line 380 It looks wired to give a summary at this place, because it is usually given at the end of the manuscript. Actually, the statements around Page 16 Line 411 are as if your conclusion.

*[This summary section is a short paragraph for $CH_4$ modeling section only, while the last conclusion section is for high-level summary and key findings*

*for the whole manuscript. We would still keep this section but make it as a sub-section of modeling section.]*

Page 18 Line 460 A few more processes not mentioned here have been presented: e.g., emission from tank bromeliads (Martinson et al., 2010) and emission from small ponds (Holgerson and Raymond, 2016).

*[We have included these new findings in the manuscript and identified them as a knowledge gap and future direction for modeling community.]*

Page 19 Line 504 I recommend adding one more (6th?) challenge. Modeling of human-natural processes such as emission from managed ponds and estuaries is important in terms of mitigation. Namely, we should consider both natural biogeochemical processes and human management effects.

*[We have added it in the revised manuscript as suggested. We appreciated the reviewer for pointing this out.]*

Page 21 Line 540Do you mean "Markov Chain Monte Carlo (MCMC)"?

*[Mistake corrected.]*

Page 25 Line 623Please correct information for Bohn et al. (2015):

*[Mistake corrected, thanks.]*

Bohn, T. J., Melton, J. R., Ito, A., Kleinen, T., Spahni, R., Stocker, B. D., Zhang, B., Zhu, X., Schroeder, R., Glagorev, M. V., Maksyutov, S., Brovkin, V., Chen, G., Denisov, S. N., Eliseev, A. V., Gallego-Sala, A., McDonald, K. C., Rawlins, M. A., Riley, W. J., Subin, Z. M., Tian, H., Zhuang, Q., and Kaplan, J. O.: WETCHIMP-WSL: Intercomparison of wetland methane

emissions over West Siberia, Biogeosciences, 12, 3321–3349, doi: 10.5194/bg-12-3321-2015, 2015.

Figure 4

Can you include the microbial community factor into the figure?

*[We have revised the figure to show several different functional groups of microbes that control the $CH_4$ processes.]*

References

Curry, C. L.: Modeling the soil consumption of atmospheric methane at the global scale, Global Biogeochem. Cycles, 21, doi:10.1029/2006GB002818, 2007.

Del Grosso, S. J., Parton, W. J., Mosier, A. R., Ojima, D. S., Potter, C. S., Borken, W., Brumme, R., Butterbach-Bahl, K., Crill, P. M., Dobbie, K., and Smith, K. A.: Gen- eral CH4 oxidation model and comparisons of CH4 oxidation in natural and managed systems, Global Biogeochem. Cycles, 14, 999-1019, 2000.

Holgerson, M. A., and Raymond, P. A.: Large contribution to inland water CO2 and CH4 emissions from very small ponds, Nature Geoscience, 9, 222–226, doi:10.1038/NGEO2654, 2016.

Martinson, G. O., Werner, F. A., Scherber, C., Conrad, R., Corre, M. D., Flessa, H., Wolf, K., Klose, M., Gradstein, S. R., and Veldkamp, E.: Methane emis- sion from tank bromeliads in neotropical forests, Nature Geoscience, 3, 766–769, doi:10.1038/ngeo980, 2010.

Matthews, E., and Fung, I.: Methane emission from natural wetlands: global distribu- tion, area, and environmental characteristics of sources, Global Biogeochem. Cycles, 1, 61-86, 1987.

Melton, J. R., Wania, R., Hadson, E. L., Poulter, B., Ringeval, B., Spahni, R., Bohn, T., Avis, C. A., Beerling, D. J., Chen, G., Eliseev, A. V., Denisov, S. N., Hopcroft, P. O., Lettenmaier, D. P., Riley, W. J., Singarayer, J. S., Subin,

Z. M., Tian, H., Zürcher, S., Brovkin, V., van Bodegom, P. M., Kleinen, T., Yu, Z. C., and Kaplan, J. O.: Present state of global wetland extent and wetland methane modelling: conclusions from a model inter-comparison project (WETCHIMP), Biogeosciences, 10, 753–788, doi:10.5194/bg- 10-753-2013, 2013.

Wania, R., Melton, J. R., Hodson, E. L., Poulter, B., Ringeval, B., Spahni, R., Avis, C. A., Chen, G., Eliseev, A. V., Hopcroft, P. O., Riley, W. J., Subin, Z. M., Tian, H., van Bodegom, P. M., Kleinen, T., Yu, Z. C., Singarayer, J. S., Zürcher, S., Lettenmaier, D. P., Beerling, D. J., Denisov, S. N., Prigent, C., Papa, F., and Kaplan, J. O.: Present state of global wetland extent and wetland methane modelling: methodology of a model inter-comparison project (WETCHIMP), Geoscientific Model Development, 6, 617–641, 10.5194/gmd-6-617-2013, 2013.

Reviewer #3

*[We really appreciate the comments, which significantly improve the manuscript in terms of clarity and organization. All detailed point-by-point responses are listed below.]*

Overall Evaluation

This manuscript presents a review of approaches used to model methane dynamics in terrestrial ecosystems in the last four decades. The review largely focuses on describing the variability in structure and mathematical descriptions of processes among 39 terrestrial methane models. Parameterization issues are touched upon in the section on environmental controls, mostly with respect to variability in Q10 (which affects temperature sensitivity of processes). The discussion makes suggestions for adding more complexity to methane models, primarily along the lines of more explicitly considering microbial processes and dynamics. The discussion finishes with identifying knowledge gaps, modeling challenges, data needs, and the need for data-model integration.

*[We appreciated the positive comments.]*

This manuscript tries to cover a lot of ground. The primary strength of the manuscript, in my opinion, is largely in the description of variability in mathematical descriptions of processes. The other aspects of the review didn't provide a lot insight in my opinion, as the issues discussed were in many cases just touched upon and were not well developed. My main concern about this manuscript is that in trying to cover a lot of ground, it covers some of that ground poorly. I think there are several issues to address to improve the review. First, I think there are some general organization issues that could be addressed to improve the manuscript. Second, there are a number of cases in the presentation of putting the "cart" before the "horse". Third, I didn't find that the description in the variability in structure (as depicted in Figure 3) was based on an objective evaluation of the 39 terrestrial models. Fourth, there a number of assertions in the manuscript that

should be presented as more open issues. Fifth, the challenge of scaling is only touched upon in the manuscript and needs to be better developed, and there is a need for some discussion of reconciliation with atmospheric data analyses. Sixth, beside the scaling/reconciliation issue, I also found several issues that need to be better developed/discussed including the modeling of ebullition, vertical representation of processes, model benchmarking, and data-model integration. Below I go into more depth on each of these issues, and finish my review with a listing of specific comments.

*[We have made a substantial revision to address the comments. Specifically, (1) we reorganized several sections to make them clearer, particularly the modeling section; (2) we clearly revised some statements to make them more consistent with the results; (3) scaling is not the key focus of this review, therefore, we did not expand writing on scaling and its reconciliation with atmospheric data analysis; yet we did emphasize that satellite data of atmospheric $CH_4$ concentration could be used for model validation; (4) we added a section for discussing presenting $CH_4$ module in ESMs.]*

Issue 1: Organizational issues in the manuscript. The manuscript starts out well, but then gradually gets more and more disorganized. There is a lot of overlap of material between some of the later sections of the manuscript that could be eliminated with a more effective organization. Perhaps consider the organization of Luo et al. (2016, Global Biogeochemical Cycles), which review soil carbon models. The organization of that paper is (1) model structure, (2) model parameterization, and (3) external forcing. I think additional in this manuscript concerns scaling and reconciliation with atmospheric data. The strength of this manuscript is that it generally does a good job of reviewing model structure, but a rather inadequate job of reviewing model parameterization, external forcing, scaling, and reconciliation issues.

*[We did remove some redundancies in different sections. This manuscript is not designed to cover model parameterization and external forcing specifically. Those two sections were discussed briefly model development perspective.]*

Issue 2: "Cart" before the "Horse" issues. There are a number of places in the manuscript where the "cart" comes before the "horse", from the perspective of this being a review paper. For example, the citation to Figure 2 on line 162 talks about the timeline for inclusion of "key mechanisms", but these mechanisms haven't been described in a general sense yet. Table 2, which contains the list of "key mechanisms" isn't cited until line 175. Even when Table 2 is cited, the general reader gets no background on these mechanisms/features of models, as it is not used beyond a simple citation at the end of a sentence. Other rough spots in the manuscript involve adequately describing terms used in the manuscript. For example, acetoclastic and hydrogenotrophic methanogenesis suddenly appears on lines 238-240 without any prior description. "Advective transport" (line 203) is also not described.

*[For the citation to Figure 2, although those processes and their representation in models have not been reviewed, yet the processes themselves have been reviewed in "primary $CH_4$ processes" section. For the acetoclastic and hydrogenotrophic methanogenesis, we have included detailed definitions at their early occurrences. The advective transport has been defined as well.]*

Issue 3: Analysis of the variability in structure. What is the basis for defining three different types of models? It seems to me that this could be done in a much more objective fashion by doing some sort of cluster analysis among the 39 models reviewed in this study. Information from Tables 1, 2, and 3 could be put into an objective cluster analysis so that we better understood what factors seem to cause models to be distinct (or not distinct) from each other.

*[We really appreciated the suggestion of doing a cluster analysis. We did a cluster analysis based on model characteristics of representation of methanogenesis processes, methanotrophy processes, transport pathways, oxygen availability, multiple soil layers etc. All 40 models could be classified into three groups, which is consistent with our previous classification. See the*

*updated Figure 3 and relevant text.]*

Issue 4: There are a number of assertions in the manuscript that have not been justified by any sort of rational analysis/argument. For example, why make a recommendation in the last sentence of section 4 (lines 196-198) on the third types of models as the means of moving forward with respect to improving reduced form models for application in Earth System Model applications? First of all, this is too early in the manuscript. Second, doesn't making this recommendation conflict with the sentence on lines 211- 212 that the optimum complexity remains to be determined? At the end of section 6 there are four recommendations for models "based on the above-mentioned needs" and a citation to Figure 4. I didn't find the previous text in section 6 as being very helpful for establishing these as the top needs. This all comes before the section 7, which talks about knowledge gaps and data needs. The arrows for benchmarking and data assimilation in Figure 4 have not been developed, and the issues of vertical trans- port/diffusion have only been touched upon. Also, the top recommendation that "the models (features?) should be embedded in an Earth System Model" seems strange to make here. The point here is that arguments have not been well enough organized and crafted to effectively make these recommendations. This sort of all gets back to issues 1 and 2 above. Finally, I can't say that I'm very fond of Figure 4 as being the synthetic figure for this manuscript – we've seen a lot of these sort of figures over the years. I suggest thinking about something that is truly synthetic based on this manuscript.

*[We have revised the manuscript to address all comments. The recommendation of the third group of model in the early section of the paper has been removed as suggested. Then it is not in conflict with later section as reviewer suggested. We added description and summary of $CH_4$ model representation in ESMs, the model classification have been done in a mathematical way -- a cluster analysis. Other writing issues have been addressed as well. The original Figure 4 is a framework showing future model development as we envisioned, it is combination of summarized and visionary framework. Although it is little lack of evidence, we do believe it will be the key direction for $CH_4$ model development and application.]*

Issue 5: The issues of scaling and reconciliation with atmospheric data. Scaling is an important issue. It does pop up several places in the manuscript as a sort of "between the lines" issue, but it really needs its own section. I also think that the issue of reconciling model applications at particular scales with data from atmospheric analyses needs to be part of the discussion.

*[Since scaling and reconciliation are not the key focus of this manuscript, we did not plan to expand that section in this revision.]*

Issue 6: Other issues. I also found several issues that need to be better developed/discussed including the modeling of ebullition, vertical representation of processes, model benchmarking, and data-model integration. For example, transport mechanisms don't even show up as key features in Table 2, although they do appear somewhat in Table 1. These issues are touched upon in several places in the manuscript, but are not really effectively dealt with in a meaningful way.

*[We have added more text and wordings to make those statements strong and solid.]*

Specific comments

Line 104-105: "contributes" is not really the right verb to use here. Just says "varies from 1 to 90%", for example.

*[We still keep "contributes" because it emphasizes the contribution of individual process to the total oxidation or production.]*

Line 106-107: I really don't know what you mean by "oxidation of atmospheric $CH_4$ contributes". Aren't all of the previous mechanisms in this paragraph ultimately oxidation of atmospheric $CH_4$, albeit in the open pore space of the soil.

*[It emphasizes the oxidation of atmospheric $CH_4$, taking up $CH_4$ from atmosphere. This process is defined to distinguish from oxidation of $CH_4$ produced from soils.]*

Line 109: Perhaps start a new paragraph after "methanotrophy.".

*[We separated it as a new paragraph.]*

Lines 109-116: There is no information for the uninitiated reader to understand how these pathways differ from each other.

*[We added one small paragraph to define different transport pathways.]*

Line 120: I think this might be the only occurrence of "wind speed" in the manuscript. What do you mean by "wind speed" as an environmental factor.

*[We revised the manuscript to have a bit more description of wind speed impacts on $CH_4$ flux.]*

Line 121: Define what you mean by "indirect" vs. "direct" environmental factors.

*[We revised the manuscript to define the direct and indirect environmental factors.]*

Line 147: I don't think Fan et al. (2013, Peatland DOS-TEM) has anything to do with the Zhuang et al. (2014) model in that it has a number of different features and to my understanding the two models do not share any code base.

*[We have confirmed with Dr. Zhaosheng Fan, and treated the DOS-TEM as another independent $CH_4$ model in the revision.]*

Line 162: As mentioned earlier, the reader needs to know more about the key mechanisms before you present/interpret Figure 2.

*[We have added definition for some key mechanisms in the manuscript.]*

Line 175: Need to make better use of Table 2 in the manuscript. As I indicated earlier, transport mechanisms need to be included in Table 2.

*[We expanded the Table 2 to include the model information on $CH_4$ transport pathway.]*

Line 213: Does use of "first group of models" refer to model types in Figure 3, or to the first set of empirical models referred to in the first paragraph of section 4.1?

*[We have revised the manuscript to be clearer on this issue. The $CH_4$ models were classified as groups, while methangoenesis was categorized as model algorithms.]*

Line 238-240: Where does the information on acetoclastic and hydrogenotropic methanogenesis appear in Table 3? Note that these production processes have not been defined for the reader.

*[We have added definitions for acetoclastic and hydrogenotropic methanogenesis in the revised manuscript.]*

Line 280: Why is Zhuang (2004) cited here in the context of immediately transporting $CH_4$? This model is primarily a monthly model with a pseudo-daily time step. This transport issue is an important temporal scaling issue, and one which should appear in a separate section on temporal scaling.

*[Thanks for pointing out this inappropriate expression. We have removed citation of Zhuang (2004), and added another model as an example.]*

Line 286: I think you should change "will likely" to "can".Line 287: I think you should change "impossible" to "not straight forward".

*[Revised as suggested.]*

Line 291: I note that ebullition is not adequately treated in this section (section 4.4).

*[We do have ebullition in the section. In the revision, we have revised the section to have more specific information for ebullition.]*

Line 292: Why is this the "final" bottleneck, or why is even referred to as a "bottleneck". Line 303: Define advective transport.

*[We revised it to bottleneck, and added definition of advective transport.]*

Line 313: I think you should change "most" to "some". Note that ebullition seems to be ignored in these three "transport" challenges. It is a dominant pathway in some systems.

*[Revised as suggested.]*

Line 319: I note that the simulation of variability in some environmental controls is not adequately treated in section 4.5 on environmental controls.

*[We have revised the section to better describe variability in environmental controls.]*

Lines 331-332: I think that this sentence needs to refer to Eq 9, 10, and 11 instead of 10, 11, and 12. Note that the third function in Eq 9 is essentially equivalent to Eq 10 in that the $Q_{10}$ can be derived from the exponent.

*[Mistake corrected.]*

Line 347: I think you mean Eqs. 13-16 instead of 12-15.

*[Mistake corrected.]*

Lines 356-367: Do any models represent pH variability in time? It would be useful to know how models represent pH variability in space.

*[We agree that pH variability is important and only few models consider dynamics of pH in soil over time and across space. Due to recent studies suggesting the importance of pH on $CH_4$ flux, it would be noteworthy to point out its importance for future model development.]*

Lines 393-394: Why is the comparison of high frequency observational data needed for future model-model inter-comparison? I think it would be most important to high quality seasonal and interannual estimates derived from observations to effectively test and compare models.

*[We have revised to reflect this point.]*

Line 405: With respect to shifts, are you referring to shifts in time or in space?

*[We have revised to clarify it is temporal shifts.]*

Line 479: What do you mean by "order 1-10". Do you mean by a "factor of 1-10"? The language could be confused for "orders of magnitude".

*[We have revised it to a factor of 1-10.]*

Luo, Y., A. Ahlstrom, S.D. Allison, N.H. Batjes, V. Brovkin, N. Carvalhais, A. Chappell, P. Ciais, E.A. Davidson, A. Finzi, K. Georgiou, B. Guenet, O. Hararuk, J.W. Harden, Y. He, F. Hopkins, L. Jiang, C. Koven, R.B. Jackson, C.D. Jones, M.J. Lara, J. Liang, A.D. McGuire, W. Parton, C. Peng, J.T. Randerson, A. Salazar, C.A. Sierra, M.J. Smith, H. Tian, K.E.O. Todd-Brown, M. Torn, K.J. van Groenigen, Y.P. Wang, T.O. West, Y. Wei, W.R. Wieder, J. Xia, X. Xu, X. Xu, and T. Zhou. 2016. Toward more realistic projections of soil carbon dynamics by Earth system models. Global Biogeochemical Cycles 30:40-56, doi:10.1002/2015GB005239.

Reviewer #4

*[We really appreciate the reviewer for the comments, which significantly improve the manuscript in terms of clarity and organization. All detailed point-by-point responses are listed below.]*

This is an excellent and timely review of the current state of process-based methane modeling. While other recent literature on particular methane models typically pro- vide some brief review in the introduction and/or discussion sections, this review paper provides a very useful level of detail for understanding where, how, and why, process- based models of methane differ. As the authors note, these current methane models often poorly reproduce observed patterns, so this is an important reflective manuscript to assess the field before moving forward. That said, I do believe that the manuscript could be improved and clarified before publication. There are several relatively minor terms and phrases that require clarification that are detailed below. On a larger point, I think that it would be helpful to provide more information about representations of $CH_4$ processes that are included within ESMs, since this is a major suggestion by the authors. They could include basic information on which models are in ESMs in Table 1, but it would also be helpful to detail plans for future representations.

*[We agree with the description of this work, and the needs for more discussion of $CH_4$ model representation in ESMs. We have added one paragraph to summarize how ESMs include $CH_4$ module; what is the likely future direction for ESMs development in terms of $CH_4$ representation.]*

Within the conclusions of their review, the authors argue that researchers should focus on the development of a fully mechanistic $CH_4$ model that accounts for all features, and can integrate data on microbial community structure and function. There is always some tradeoff with model complexity and functionality, and I would be more convinced by the authors' conclusions that a more complex mechanistic model should be developed with all components if there was some evidence that this improves simulations over

simpler representations. And furthermore, how can the increasing number of plot- to ecosystem-scale measurements of net $CH_4$ flux be used to constrain such a complex model, except for validation? This type of very complex model would even more so require the aggregation of experimental data on microbial ecophysiology that can be used to parameterize and develop robust uncertainties for these processes, and the authors appropriately note that much of this experimental work is yet to be done. It would be helpful if the authors provided some context for understanding how much data exist to constrain these individual $CH_4$ processes (a handful of experiments, or potentially hundreds?) and within which ecosystems. Within the section on model-data integration, I also think that it would be useful for the authors to provide more specific detail regarding ways to integrate these different data types (from net $CH_4$ flux data to process-based experimental data).

*[We have added discussion about the tradeoff of developing a more mechanistic model and a simple empirical model. Meanwhile, the classification of empirical model and process-based model has been expanded. Meanwhile, we totally agreed that constraining mechanistically model is really challenging, yet it is becoming more and more applicable as the scientific community is expanding measurements of $CH_4$ flux and processes, as well as developing new model optimization algorithms. For example, SPRUCE, NGEE-Arctic and NGEE-Tropic projects within DOE are taking this intensive measurements and integration with models. A new model optimization algorithm has been developed associated with CLM framework and ALM framework, we believe the mechanistic models will be more powerful in near future along with these lines of advancements.]*

Line by line comments follow:

L102-109: I'm confused about the reference number for the percentages: is it the percent of total carbon respiration? Or percent of total methane produced?

*[Those percentage numbers emphasize specific processes to the single function of $CH_4$ cycling. For example, acetoclastic methanogenesis contributes to ~60-100% to the total $CH_4$ production.]*

L235: You should also consider citing Matthews & Fung (1987) in this history: Matthews, E., and I. Fung (1987), Methane emission from natural wetlands: Global distribution, area, and environmental characteristics of sources, Global Biogeochem. Cycles, 1(1), 61–86, doi:10.1029/GB001i001p00061.

*[We totally agree that the pioneering work by Matthews and Fung is important and should be cited in the manuscript. We have cited it in the revised manuscript.]*

Table 1: Since the table is already large, I think that it would be useful to add which models are within ESMs (and if so, which ESM) and which models were developed for particular regions/species (rice, Arctic, etc.).

*[We appreciated the reviewer for pointing out this issue. The information of which $CH_4$ models are embedded in ESMs has been summarized in the Table.]*

L280-295: I think it would be helpful to add a bit more context for how and why these CH4 models are added into ESMs. The authors recommend that the third group be the focus to understand potential for reduction into ESM models, but what does it take to reduce a CH4 model into an ESM?

*[We have added texts to emphasize the importance of representing $CH_4$ module in ESMs.]*

L315-330: This section is a bit hard to follow with respect to what exactly the differences are here among the models. I think that it would be useful to restructure this with a bit more of an introduction (like the environmental controls section) about the differences among the four distinct classes of substrate representation, with explicit list of the four classes before listing

which model is in each class.

*[This section has been re-organized little bit for clarity purpose.]*

L345: I'm not sure what the authors mean by "dramatic bias" caused by a lack of representation, and this should be clarified.

*[We have revised the sentence to clearly reflect the importance of representing these two mechanisms; the bias in surface $CH_4$ flux will likely be biased if we do not represent these two mechanisms. Studies have confirmed that the surface layer and bottom layer have different mechanisms dominated $CH_4$ production (McCalley et al., 2014), therefore, if we do not consider two mechanisms, we will not be able to simulate this shift and likely the surface fluxes caused by this function shift in response to environmental change.*

*McCalley, C. K., Woodcroft, B. J., Hodgkins, S. B., Wehr, R. A., Kim, E.-H., Mondav, R., Crill, P. M., Chanton, J. P., Rich, V. I., Tyson, G. W., and Saleska, S. R.: Methane dynamics regulated by microbial community response to permafrost thaw, Nature, 514, 478-481, 2014.]*

L363: It's hard to follow the many different categories that the authors are creating, and I'm not completely sure which category three refers to as described here.

*[We have revised the manuscript. The three groups of $CH_4$ models are remained, while we changed four groups of methanogenesis to four modeling algorithms for methanogenesis. The classification of three groups of $CH_4$ models have been demonstrated with a cluster analysis as suggested by another reviewer (new Figure 3).]*

L370: It would be helpful to provide a bit more context for why Michaelis-Menten representation fails for multi-substrate, multi-consumer networks. Is it purely an equifinality problem?

*[We rewrote this sentence to acknowledge the new approach developed by Riley's group. The ECA approach might be good for multi-substrate, multi-consumer biogeochemistry reaction network. We added a short description in this aspect.]*

L398: Unclear what "reported these individual processes" is referring to.L479: I'm not sure what the "high range" refers to within this context.

*[L398 emphasizes the individual processes discussed in previous section. While L479 primarily focuses on processes caused hot spot and hot moments in $CH_4$ flux. In the revised manuscript, we revised those two sentences for clarity purpose.]*

L567: Unclear what is meant by "integrative tool" . . . for integrative assessment?

*[We used "integrative tool" to emphasize that the model can be used to integrate multiple sources of data to reach a better understanding of the system and better budget quantification.]*

Reviewer #5

*[We really appreciate the reviewer for the comments, which significantly improve the manuscript in terms of clarity and organization. All detailed point-by-point responses are listed below.]*

The manuscript by Xu et al. reviews the past four decades of modeling methane emissions from terrestrial ecosystems. The authors provide a timeline and structure for assessing both the level of detail in terms of the processes represented and also in terms of how the processes are represented. Overall, the authors do a very nice job of comprehensively summarizing the current state of art in methane modeling and tracing the history of model development over the past four decades.

*[We appreciate the positive comments.]*

My main comments are : 1. The authors categorize the representation of processes into empirical to mechanistic approaches. This is rather subjective and it would be very helpful for the reader to have a section (1-2 paragraphs) describing how the authors define these terms. For example, even some of the mechanistic representation of processes rely on empirical response functions, and are thus only semi-mechanistic. In an ideal setting, what would be the definition of a purely mechanistic modeling approach?

*[We totally agree that separation of mechanistic and empirical is rather arbitrary, while it does help understand the model representation of $CH_4$ processes. In this revised manuscript, we provideddetailed description to show how to define the empirical and mechanistic models in terms of modeling $CH_4$ dynamics.]*

2. Some of the descriptions of the processes are fairly vague. For example, even the description of methanogenesis is abbreviated to just mentioning "acetoclastic and hydrogenotrophic methanogenesis". Given that the authors

are trying to emphasize a more mechanistic modeling approach, increasing the level of detail for each process would be helpful.

*[We agree that it is important to have more detailed description of two methanogenesis processes. Yet the processes themselves have been reviewed, while for all the present $CH_4$ model, only few models simulate acetoclastic and hydrogenotrophic methanogenesis; which are not sufficient for a detailed review section.]*

3. The discussion on substrate is particularly useful because most methane models do not consider this explicitly. Given the rise of atmospheric CO2, addressing how substrate has changed due to CO2 interactions, and what this means for modeling approaches and methane emissions is necessary to be mentioned.

*[We added a short paragraph to discuss the potential impacts of elevated $CO_2$ and substrate on $CH_4$ emission.]*

4. Lastly, in the discussion for data needs, the list and ideas for integration within models is also very helpful. However, some discussion of the benchmark targets that the modeling community should aim for, and how to handle the uncertainties in bench- marks, would be very useful.

*[We have added one small paragraph to summarize the benchmarking targets of the benchmarking system and uncertainties in benchmark.]*

---

## Author Comment (AC7) · 20 May 2016

L370: It would be helpful to provide a bit more context for why Michaelis-Menten representation fails for multi-substrate, multi-consumer networks. Is it purely an equifinality problem?

[The Equilibrium Chemistry Approximation (ECA) was developed to represent chemical kinetics when there are multiple substrates and multiple competitors. These conditions are inconsistent with the original derivation of the Michaelis-Menten (MM) kinetics by Briggs and Haldane, and subsequent investigations over the past decades have indicated many cases where MM kinetics are inaccurate. Since the purpose of this paper is not to give detailed explanations of the various process representations in the models, but rather to indicate briefly their various components, we added a sentence to the

manuscript and refer the readers to several publications applying this concept (Tang and Riley 2013, Tang 2016, Zhu et al. 2016).]

---

## Author Response (AR2)

This manuscript represents a considerable volume of work, and I congratulate you for completing such a comprehensive literature review. My hope is that this publication will serve as an important reference for methane modelling, measuring and data integration activities, facilitating the development of the next generation of models.

The text reads well, and I think you managed to incorporate the referees' comments well. I have a few further (mostly minor) edits below, which I would like to ask you to change on the latest version, which will then be ready for final acceptance.

[Responses: Thank you very much for the positive comments. These detailed editorial comments are valuable as well. We have further revised the manuscript to address all those comments. In addition, we have re-formatted the section of *5. Challenges for Developing Mechanistic CH$_4$ Models*. The italic texts for each sub-section have been reformatted as sub-heading. This has been conceived in last revision.]

Manuscript comments:

There is a tendency to use many citations, when a point could be supported by a section of references. For example, do you need all 12 references to make the point that ecosystem modelling is a broadly used tool (lines 42-47)? It is within the nature of a review paper that you include most of the literature written on the subject, but in the introduction, I think that a more selective approach to introduce the background to your manuscript would work better.

[Responses: We have revised and reduced the number of citations in this paragraph.]

Line 81: Add "further" after "This review".

[Responses: Revised as suggested.]

Line 95/96: Do you need citations here?

[Responses: Thanks for the comments. The citations we put in this sentences are all review papers on methane processes, we feel that these reviewer covers most of the key CH$_4$ processes being studied and documents. Therefore, we would still keep them in this sentence.]

Line 67: Figure 2 does not demonstrate an increase in the number of models

[Responses: Thanks for the comments. We corrected it to Figure 1.]

Line 212: Delete "of them".

[Responses: Revised as suggested.]

Line 340: Delete "directly". ("Direct incorporation" into models is confusing, when you talk also of direct and indirect drivers).

[Responses: Revised as suggested.]

Line 357: "an important", rather than "another important".

[Responses: Revised as suggested.]

Line 365: "moisture effects", rather than "moisture's effects".

[Responses: Mistake corrected.]

Line 375: Delete "is another important factor that".

[Responses: Revised as suggested.]

Line 410: Add "The" before "IAP-RAP model".

[Responses: Mistake corrected.]

Line 418: "changes in CH4 flux have…", rather than "has".

[Responses: Mistake corrected.]

Line 430: Add "or" before "anaerobic".

[Responses: Thanks for the comments. We added "and" rather than "or" because those two gaps both exist.]

Lines 486-488: This is confusing, and needs to be rephrased. I suggest an alternative here, but please check carefully if this expresses what you want to say here: "Iron and sulfate biogeochemistry has so far been modelled implicitly by only a few models, as mechanisms are as yet poorly understood, and there is a paucity of data. Accordingly, these processes have not been incorporated into recently developed models, and a more explicit inclusion, based on improved biogeochemical understanding, will hopefully be achieved in the long term.

[Responses: Thanks for the suggested revision. It looks perfect and has been used in the revisions.]

Line 498: Comma after "identified".

[Responses: Mistake corrected.]

Lines 503/504: Rephrase to:"One well-known mechanism is aerobic…"

[Responses: Revised as suggested.]

Lines 508/509: Please rephrase to: "The second mechanism is CH4 production by fungi (Lenhart et al 2012)."

[Responses: Revised as suggested.]

Line 614: Reference to pers. Communications not needed here.

[Responses: Revised as suggested.]

Page 55: Figure legends:
Figure 1: Delete "The" at beginning of sentence. Replace "at decadal scale" (which implies a sale over which models are applied) by "over recent decades".

[Responses: Thank you for the suggestions. It has been corrected.]

Figure 3: Do you mean "three", or "the"?

[Responses: It is "three". Mistake corrected.]

Page 57 (Figure 2): This is poorly formatted. The x-axis tick labels are not fully represented, and the overall size could be bigger. Please also remove dashed background lines in the chart area. Rather than open and hatched columns, I suggest solid fill for all data series, with white, grey and black as fill colours.

[Responses: Revised as suggested; the x-axis tick labels have been fully represented, the color coding for the bars have been updated as well.]

[revised manuscript text omitted]

$O_2$
$CH_4$
$CO_2$
$CH_4 + O_2 \rightarrow CO_2$

water    DOC
Soil (SOM) → DOC
ACE
$CO_2$
$H_2$
$CH_4$
$NO_3^-$, etc. electron acceptor

Fig. 5